# Abrupt Learning in Transformers:
# A Case Study on Matrix Completion

**Pulkit Gopalani**
University of Michigan
gopalani@umich.edu

**Ekdeep Singh Lubana**
Harvard University
ekdeeplubana@fas.harvard.edu

**Wei Hu**
University of Michigan
vvh@umich.edu

## Abstract

Recent analysis on the training dynamics of Transformers has unveiled an interesting characteristic: the training loss plateaus for a significant number of training steps, and then suddenly (and sharply) drops to near–optimal values. To understand this phenomenon in depth, we formulate the low-rank matrix completion problem as a masked language modeling (MLM) task, and show that it is possible to train a BERT model to solve this task to low error. Furthermore, the loss curve shows a plateau early in training followed by a sudden drop to near-optimal values, despite no changes in the training procedure or hyper-parameters. To gain interpretability insights into this sudden drop, we examine the model's predictions, attention heads, and hidden states before and after this transition. Concretely, we observe that (a) the model transitions from simply copying the masked input to accurately predicting the masked entries; (b) the attention heads transition to interpretable patterns relevant to the task; and (c) the embeddings and hidden states encode information relevant to the problem. We also analyze the training dynamics of individual model components to understand the sudden drop in loss.

## 1 Introduction

Large Language Models (LLMs) have revolutionized the field of natural language processing (NLP). However, there are still gaps in our understanding of these models, leading to challenges in controlling their behavior. As a pertinent example, the training of these models appears to demonstrate *sudden* improvements in metrics correlated with various capabilities [8], prompting questions about whether learning of a given capability can be predicted by tracking predefined progress measures and *why* such sudden changes occur. If undesirable capabilities can suddenly 'emerge' (despite any explicit supervision for them) [16], such sudden changes can be a challenge for AI regulation [21].

To better understand such sudden changes during model training, this work investigates training BERT [12] on the classical mathematical task of low-rank matrix completion (LRMC) [6]. Making an analogy with masked language modeling (MLM), where sudden learning of syntactical structures was recently demonstrated [8], we argue matrix completion captures the core aspect of this learning problem (Fig. 1): given some relevant context (observed tokens), fill the missing elements (masked tokens). Specifically, we assume access to a matrix with some fraction of its entries missing, and would like to complete the missing entries of this matrix assuming the ground truth matrix is low-rank. We find that despite being a simplified abstraction of MLM, this setting already demonstrates *a sharp* decrease in loss as the model undergoes training (Fig. 1 (B)), preceded by a loss plateau for a significant number of training steps (akin to Chen et al. [8]). The simplicity of our setting further affords us interpretability, as we find that the point of sudden drop coincides with a precise change in how the model solves the task—we call this change an *algorithmic transition*. Specifically, we show that the pre–transition model simply copies the input (predicting 0 at masked positions), while the post–transition model accurately predicts missing values at masked positions. To perform the latter,

38th Conference on Neural Information Processing Systems (NeurIPS 2024).

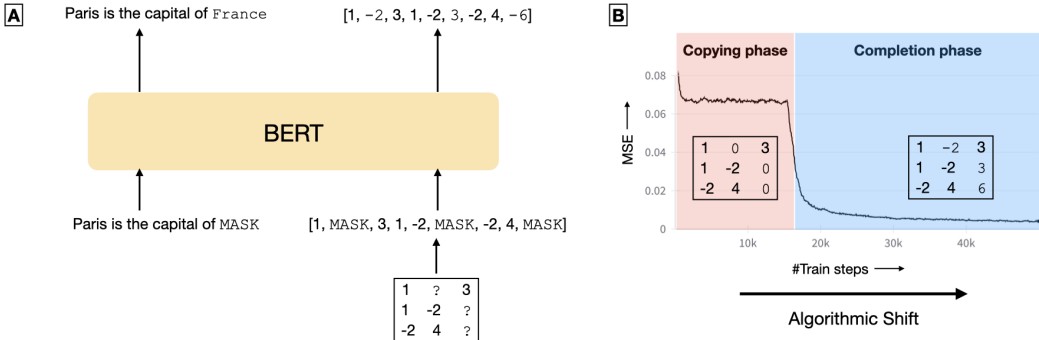

Figure 1: (**A**) **Matrix completion using BERT**. Similar to completing missing words in an English sentence in MLM, we complete missing entries in a masked low–rank matrix. (**B**) **Sudden drop in loss**. During training, the model undergoes an *algorithmic shift* marked by a sharp decrease in mean–squared–error (MSE) loss. Here, the model shifts from simply copying the input (*copying phase*) to computing missing entries accurately (*completion phase*).

distinctive changes occur in the model's attention heads during the period of sudden drop, wherein the model learns to identify relevant positional information to combine various elements in the input matrix and compute missing entries for matrix completion. We perform a range of interventions on the input, model (before and after the transition), and training process to further understand this phenomenon, leading to the following observations.

- **Pre–transition: Copying the Input Matrix**    Before the transition, the model is simply copying the input matrix both at observed entries as well as missing entries, predicted value for missing entries being nearly $0$. The attention maps at this stage do not correspond to a particularly interpretable structure, and contribute little to the model output.

- **Post–Transition: Computing Missing Entries**    After the transition, the model accurately completes the missing entries, while still copying observed entries. The attention maps at this stage clearly demonstrate that the model 'attends' to relevant tokens in the input, and the attention layers are crucial for accurate matrix completion. Interestingly, the post–transition model can outperform the classical nuclear norm minimization algorithm for matrix completion, suggesting that it does not simply recover this algorithm.

- **Model Components and Sudden Drop**    We analyze the training dynamics of individual components, keeping other components fixed to their final values. We find that different components converge to their optimal values at quite different points during this training.

## 2    Preliminaries

### 2.1    Problem Setup

**MLM and LRMC**    In masked language modeling (MLM), a fraction of tokens in the input sequence are masked out and the model is required to predict the correct token for those masked entries. In this setup, the model has access to both the tokens before and after the current token for computing the prediction. Low-rank matrix completion has a similar structure: given a matrix (assumed low–rank) with a fraction of its elements available, the goal is to predict missing entries. For a matrix $X \in \mathbb{R}^{n \times n}$, denote its observed entries by the set $\Omega \subset [n] \times [n]$, and the set of missing entries $\Omega^{\mathsf{c}} = [n] \times [n] \setminus \Omega$. Formally, the problem is

$$\min_{U} \ \operatorname{rank}(U) \qquad \text{s.t. } U_{ij} = X_{ij} \ \ \forall (i,j) \in \Omega.$$

Importantly, both problems (MLM and LRMC) have the same goal—predict the missing entries in the input, i.e., either the language tokens (MLM) or matrix elements (LRMC).

**Matrix Completion using BERT**    BERT [12] is an encoder-only Transformer architecture used widely for MLM. For an input sequence of tokens $[t_1, t_2, \ldots, t_L]$, the output is a sequence of

$D-$dimensional 'hidden states' $[e_1, \ldots, e_L]^\top \in \mathbb{R}^{L \times D}$, that is used for prediction. We train a BERT model $\mathsf{TF}_\theta$ to predict missing entries in a low–rank masked matrix $\tilde{X}$. For model output $\hat{X} := \mathsf{TF}_\theta(\tilde{X}) \in \mathbb{R}^{n \times n}$, the training objective $L := L(\theta)$ is the mean-squared-error (MSE) loss over all entries,

$$L(\theta) = \frac{1}{n^2} \sum_{i,j=1}^{n} (X_{ij} - \hat{X}_{ij})^2.$$

In our experiments, data for matrix completion is generated as

$$X = UV^\top; \quad U, V \in \mathbb{R}^{n \times r}, \;\; U_{ij}, V_{ij} \overset{\text{iid}}{\sim} \text{Unif}[-1, 1] \;\; \forall i, j \in [n] \times [r]$$

so that $X$ has rank at most $r$. To mask entries at random, we sample binary matrices $M \in \{0, 1\}^{n \times n}$ such that $M_{ij} = 0$ with probability $p_{\text{mask}}$, and 1 otherwise; that is, $\Omega = \{(i, j) \mid M_{ij} = 1\}$.

**Nuclear norm minimization**   Nuclear norm minimization [6] is a widely used convex optimization approach to LRMC; for completeness, we compare our trained models to this approach. Since rank is not a convex function of the matrix, one modifies the low rank completion problem by defining the nuclear norm $\|U\|_*$, i.e., sum of singular values of a matrix $U$. The overall optimization problem is as follows.

$$\min_U \|U\|_* \qquad \text{s.t. } U_{ij} = X_{ij} \;\; \forall (i, j) \in \Omega. \tag{1}$$

## 2.2   Experiments

**Training**   We use a 4–layer, 8–head BERT model [40] for $7 \times 7$ (rank$-2$) matrices, with 'absolute' positional embeddings, no token–type embeddings, and no dropout. We fix $p_{\text{mask}} = 0.3$ for training, and 256 matrices are sampled as training data at each step (in an 'online' training setup). We use Adam optimizer with constant step size $1\mathrm{e}{-4}$ for 50000 steps, without weight decay or warmup. In addition to $L$, we track MSE over observed and masked entries,

$$L_{obs} = \frac{1}{|\Omega|} \sum_{(i,j) \in \Omega} (X_{ij} - \hat{X}_{ij})^2, \quad \text{and} \quad L_{mask} = \frac{1}{|\Omega^{\mathsf{c}}|} \sum_{(i,j) \in \Omega^{\mathsf{c}}} (X_{ij} - \hat{X}_{ij})^2.$$

Please see Appendix D for details on tokenizing matrices and other experimental details. Code is available at this `https://github.com/pulkitgopalani/tf-matcomp`.

**Compute Resources**   For $7 \times 7$ matrices (training and testing), we used a single {V100 / A100 / L40S} GPU. A single {A40 / A100 / L40S} GPU was used for matrices of order 10, 12, 15.

## 3   Sudden Drop in Loss

In our training setup, the model converges to a final MSE of approximately $4\mathrm{e}{-3}$ – that is, it can solve matrix completion well (as in Fig. 3, this MSE is lower than nuclear norm minimization). Fig. 2 demonstrates the loss dynamics over the course of training the model on this task.

Interestingly, we observe a sudden drop in training loss at approximately step 15000. This sudden drop in loss is reminiscent of *phase transitions* in physical systems, that are characterized by sudden observable changes in the system on continuous variation of some parameter (here equivalent to the number of training steps). Motivated by this similarity, we analyse the 'pre–shift' model at step 4000, and 'post–shift' model

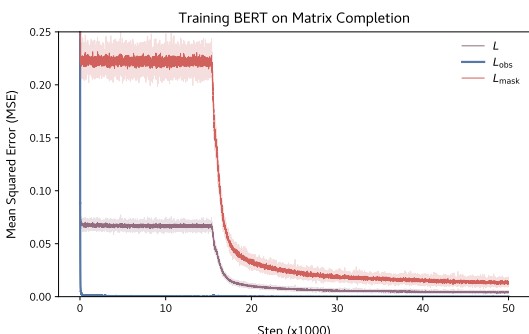

Figure 2: Sharp reduction in training loss.

at the end of training, i.e., step 50000 to understand model properties and sudden drop in loss.

### 3.1 Before the Algorithmic Shift – Copying Phase

Since the value of $L_{obs}$ remains quite low in the first phase of model training (Fig. 2), we ask: *what algorithm does the model use for predicting matrix entries in this phase?*

We find that the model learns to copy the input verbatim in the first phase (with output 0 for missing entries), verified through token interventions (Sec. 3.1.1) and by investigating the contribution of attention heads (Sec. 3.1.2) towards the output.

#### 3.1.1 Verifying Copying via Token Intervention

To rigorously verify that the pre-shift model indeed copies the input, we replace the masked elements in the $7 \times 7$, rank-2 input by the token corresponding to some $m \in \mathbb{R}$. For such input, we would like to see whether the model implements copying and outputs $m$ at the masked positions. In this setup for model output $\hat{X}$, MSE at observed positions is $L_{obs}$, and for masked positions the MSE is defined as

$$L'_{mask} = \frac{1}{|\Omega^c|} \sum_{(i,j) \in \Omega^c} (\hat{X}_{ij} - m)^2.$$

$L_{obs}$ and $L'_{mask}$ for this experiment averaged over $512$ samples are compiled in Table 1 (Appendix A). The small loss values confirm that model output matches the ground truth at observed positions, while at masked positions it outputs a value nearly equal to $m$. When the mask token is $\text{MASK}$ (i.e., no replacement), we set $m = 0$, indicating that the model outputs 0 at the masked locations.

To generalize this observation to OOD matrices, we sample uniform random $7 \times 7$ matrices for input; i.e., all entries in the matrix are i.i.d. uniformly in $[-1, 1]$. Importantly, these matrices do not necessarily have a low–rank structure. With these matrices as input to the same pre–shift model as before, we find that model still copies the input (Table 1). This confirms that the model is indeed not 'computing' any entries in the sense of low–rank matrix completion, and simply copies all entries, masked or observed.

#### 3.1.2 Attention Heads – Mostly Inconsequential

Attention heads at this stage (Fig. 22a) do not appear to attend to tokens in an interpretable manner. Since the model is copying the input, and does not need to combine different tokens, Attention heads should not affect the model output at this stage. To confirm that this is indeed the case, we do the following tests.

**Uniform Ablation** Uniform ablation entails replacing the softmax probabilities in an $n \times n$ attention head by $1/n^2$ for all elements i.e. 'force' the model to equally attend to all tokens (Sec. 4.6, [22]). On such an intervention in our case, there is negligible change in MSE at both observed and masked positions. Averaged over 256 samples, $L_{obs} = 3.4\text{e}{-}4$ and $L_{mask} = 0.2236$ when using all attention heads; whereas, on ablating all heads, these values are $3.2\text{e}{-}4$ and $0.2236$ respectively. *The negligible change in MSE supports the hypothesis that attention does not contribute to the model output at this stage.*

**Model Switching** In the extreme case, what if we replace the model weights for some component to check for changes to the output? In model switching, we 'transplant' the attention key, query and value weights in the pre–shift model to those from the post-shift model. Averaged over 256 samples, $L_{obs}$ is 5e–3, that is similar to the optimal total MSE ($L$) obtained at the end of training, while $L_{mask} = 0.2246$, similar to the values obtained without such replacement. *This shows that replacing the pre–shift attention weights by the optimal ones does not significantly affect $L_{obs}, L_{mask}$ – highlighting that attention layers have little effect on the model output at this stage.*

### 3.2 After the Algorithmic Shift – Matrix Completion Phase

In this section, we focus on the model properties in the post–shift phase (specifically, at the end of training at 50000 steps). Since $L$ are near–optimal in this setting, we ask : *What algorithm is the model using for completing missing entries? For example, is it implementing the classical nuclear-norm minimization algorithm?* For the second question, we show below that the BERT model is *not* implicitly implementing nuclear norm minimization for completing missing entries in the input.

**Nuclear Norm Minimization** We use CVXPY [13] to solve low–rank matrix completion using nuclear–norm minimization at various levels of $p_{\mathrm{mask}}$, comparing it to the output of a BERT model trained on $p_{\mathrm{mask}} = 0.3$. We find that BERT performs better than nuclear norm minimization with respect to MSE; at the same time, the nuclear norm of BERT solution is larger (Fig. 3).

To verify if the model implicitly optimizes a different objective for nuclear norm minimization, we also compare to the regularized version of the above problem ($\lambda > 0$),

$$\min_U \left[ \frac{1}{|\Omega|} \sum_{(i,j)\in\Omega} (U_{ij} - X_{ij})^2 + \lambda \|U\|_* \right]$$

We find that this is not the case, as for various values of $\lambda$, BERT still outperforms regularized MSE minimization w.r.t. MSE (Appendix B). *This confirms that the model is not implementing nuclear norm minimization as its algorithm for computing missing entries.*

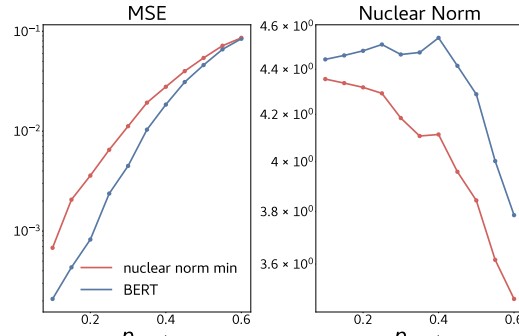

Figure 3: **BERT v. Nuclear Norm Minimization**. Comparing our model (trained with $p_{\mathrm{mask}} = 0.3$) and nuclear norm minimization on the matrix completion task at various levels of $p_{\mathrm{mask}}$. The difference in MSE and nuclear norm of solutions obtained using these two approaches indicates that BERT is not implicitly doing nuclear norm minimization to complete missing entries.

We now move to an interpretability based analysis of the model behavior, to attempt to extract useful signal about the implemented algorithm, analysing model behavior for observed and missing entries separately in the following sections.

### 3.2.1 Observed Entries

**Uniform Ablation** As in Sec. 3.1.2, to quantify the effect of attention heads at this stage, we uniformly ablate *all* attention heads in the post-shift model. Averaged over 256 samples, this leads to $L_{obs} = 9.2\mathrm{e}{-}5$ without ablation, and $3.7\mathrm{e}{-}3$ with ablation (close to the value of $L$ at the end of training). However, $L_{mask}$ increases from $0.0128$ to $0.2183$, approximately the value of $L_{mask}$ in the loss plateau before sudden drop. *This difference in effect of ablating attention heads confirms that they are much more important for predicting missing entries than for observed entries.*

**Model Switching** We repeat the model switching experiments from Sec. 3.1.2 in the reverse direction i.e. 'transplant' attention key, query, value weights from pre–shift model to the post–shift model. Note that this direction of weight switching is stronger, in the sense that the learnt information in attention layers is removed. We find that on this modification, $L_{obs} = 9.5\mathrm{e}{-}4$ averaged over 256 samples; that is, the observed loss is still not too large. *This test confirms that the prediction at observed entries is not substantially affected by the attention layers.*

**Position Sensitivity** Finally, since the attention mechanism crucially depends on token positions, we intervene on this component of the model by randomly permuting its positional embeddings. Formally, the embedding originally for position $i$ in the input now represents position $\pi(i)$ for some random permutation $\pi : [n^2] \to [n^2]$. Averaged over 256 samples, $L_{obs} = 2.4\mathrm{e}{-}4$, whereas $L_{mask} = 0.5687$, indicating that the observed positions are negligibly affected compared to masked positions due to this intervention.

These results support our 'sub–algorithm' hypothesis; (a) since positional information is intuitively not required for the copying sub-algorithm, $L_{obs}$ remains low; and (b) $L_{mask}$ increases significantly, demonstrating that removing positional information is detrimental to accurately computing missing entries.

### 3.2.2 Missing Entries

To confirm that attention heads causally affect the model output for missing entries, in addition to uniform ablations, we perform *causal interventions* (activation patching) [42] on the hidden states just after the attention heads. This involves replacing the hidden state after an attention head for input

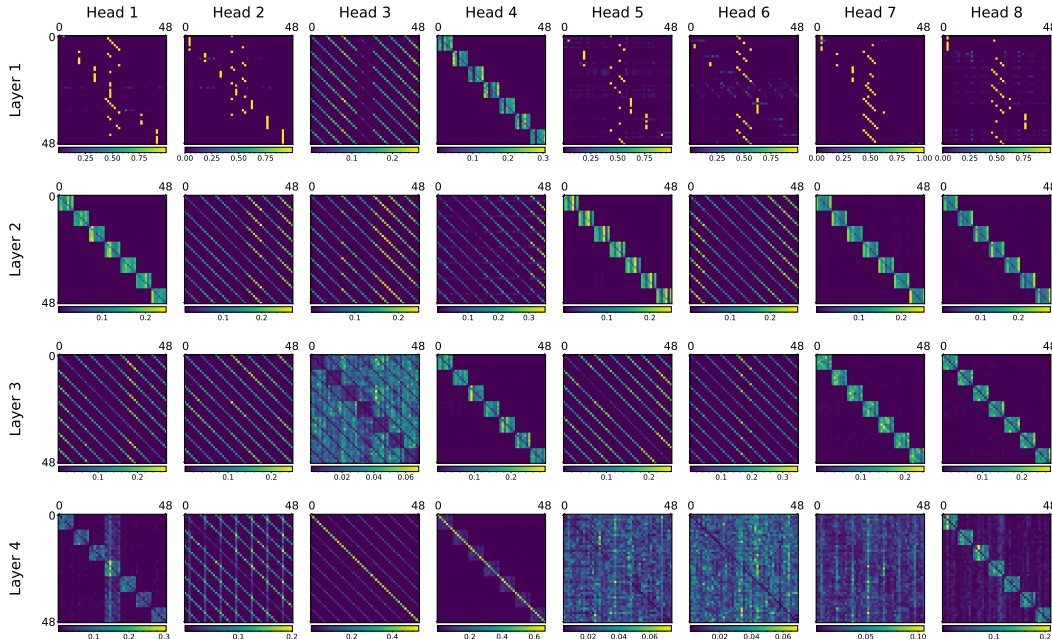

Figure 4: **Attention heads in post–shift model attend to specific positions.** For example, (Layer 2, Head 1) attends to elements in the same *row* as the query element, and (Layer 2, Head 2) attends to elements in the same *column* as the query element. (These attention matrices are an average over multiple independent matrix and mask samples.)

$A$ with the hidden state obtained at the same attention head, but for a different input $A'$. Ideally, if that head is causally relevant to the output, then such an intervention should steer the model towards the output for $A'$, instead of $A$. We find in our case that for $A = X$ and $A' = -X$, such an intervention on all attention heads clearly steers the model output at missing entries towards $-X$ (more details in Appendix F).

**Structure in Attention Heads**    Denote attention head $H$ in layer $L$ by the tuple $(L, H)$. We can group the attention heads depending on the specific regions of the input matrix they attend to,

1. **[Row Head]**   same row as the query element – 'block–diagonal' patterns, e.g. $(2, 1)$;

2. **[Column Head]**   same column as query element – 'off–diagonal' patterns e.g. $(2, 2)$; and

3. **[Identity Head]**   query element itself – 'diagonal' patterns in the last layer, e.g. $(4, 3)$.

There are also some other attention heads that do not neatly fit into either of these 3 categories—for example, all heads in layer 1 except $(1,3)$, $(1,4)$; $(3,3)$; $(4,2)$, $(4,5–7)$. In this context, we note that uniformly ablating heads $(3,3)$, $(4,2)$, $(4,5–7)$ gives $L_{obs} = 9.36e{-}5$, $L_{mask} = 0.01575$ compared to $L_{obs} = 9.44e{-}5$, $L_{mask} = 0.01428$ without ablation, i.e. these uninterpretable heads do not significantly affect the output.

**Attention Heads with 'Structured Masking'**    Since the maps in Fig. 4 are averages over multiple random masks and input matrices, it is difficult to derive more fine–grained insights into the model computation. To address this, we generate inputs with specific mask structure, see for example Fig. 5. This implies that while averaging the attention probabilities over different input matrices, the mask i.e. $\Omega^{c}$ remains the same. This step helps us highlight how an attention head attends to input elements based on the element being masked or observed. From the results in Fig. 5, we find clear evidence that different attention heads focus on specific parts of the input. For instance,

1. **[Masked–Row Head]** $(2, 1)$, $(3,4)$ and $(4,8)$ are mainly active only at the masked rows, and therein attends to the only observed position in those rows.

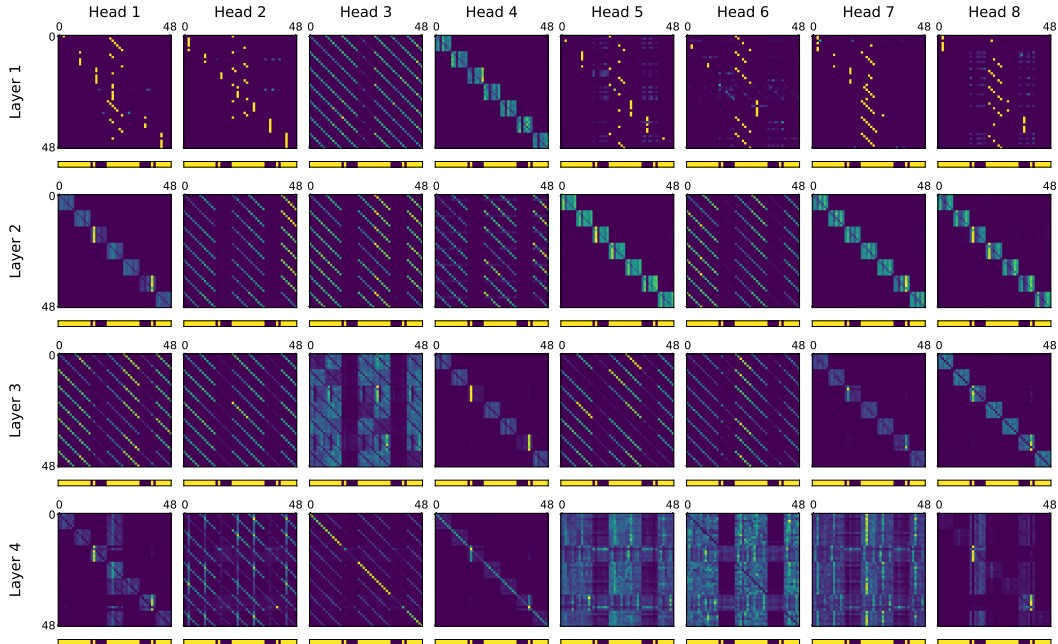

Figure 5: **Attention heads with specific mask structure in inputs**. We can derive fine-grained insights about the functions of individual heads in this setup by using a specific mask structure for all input matrices. (Mask appended below each plot, blue denotes missing entries). For example, multiple attention heads like (Layer 2, Head 2) have negligible attention weight at missing positions in the input matrix, implying that these heads attend only to observed entries in the column of the query element. Further, (Layer 2 Head 1) and similar heads have larger attention weights for the rows with missing entries, and in those rows they attend to the sole observed element.

2. **[Observed–Copy Head]** (4,3) and (4,4) correspond roughly to an identity map, slightly deviating in the masked rows. In these cases, again the maximal attention score corresponds to the only observed position in these rows.

3. **[Mask–Ignore Heads]** Further, there are multiple 'parallel off-diagonal' heads that completely ignore the masked rows for their computation. These heads include (2,2–4), (2,6); (3,2), (3,3), (3,5). Additionally, there are also attention heads like (3,1), (3,6) that attend to only the observed element of each masked row. Collectively these heads act as 'mask-ignore' heads, attending to only observed entries, and using this information to compute missing entries.

4. **[Longest–Contiguous–Column Heads]** There also exist attention heads that respond systematically to changes in the mask. For example, consider attention heads (2, 5), (2, 7), (2, 8) in Fig. 23. For each row, these heads attend to the element in the 6th and 2nd column respectively for part (a) and (b). On a closer look, the connecting link between these two mask patterns is that, the longest contiguous unmasked column is exactly the column that these heads attend to. We hypothesize that this information is somehow used by the model in its inner computation for masked entries.

5. **[Input–Processing Heads]** Finally, Heads (1,1–2), (1, 5–8) do not fall in any of the categories above . These heads are mostly static across different mask / input variations (for example, comparing Fig 4 and 5), and the patterns suggest that these heads almost exclusively focus on the middle row of the input matrix and some other elements. A possible function of these heads is to process positional and token embeddings (input to the first layer) so that this information can be used appropriately in the subsequent layers.

To quantitatively assess the effect of these attention heads on the model output, we also perform uniform ablations on each sub–group separately (Appendix L), and find that the groups significantly affect the output, to varying degrees depending on the specific group.

**Probing** We probe for properties of the input matrix in the hidden states of the model, to concretely determine how the model computes the output. We use our 12–layer model in this case, to enhance contrast between probing results in different layers.

Specifically, for every element in the input, we fit a linear probe [3] on its hidden state after a given layer, mapping the hidden state to the $n-$dimensional masked row that this element belongs to (missing entries are replaced by 0). That is, element at position $(i, j)$ maps to the $7-$dimensional vector $\tilde{X}_i$. The results for this experiment in Fig. 6 demonstrate that the hidden states at layer 3 and 4 in the model correlate quite strongly with the probe target, compared to other layers. *This result suggests that the model tracks input information in its intermediate layers and possibly uses it for computing missing entries.*

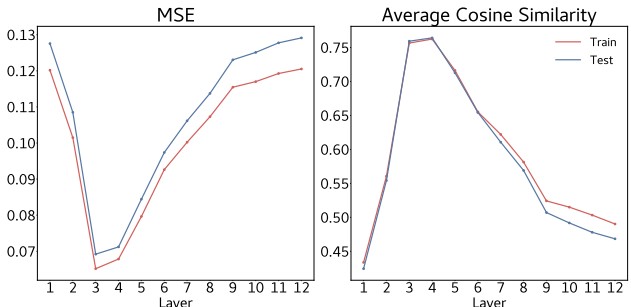

Figure 6: **Hidden states encode input information**. We probe for the row of each element in the masked matrix input, replacing missing entries by 0. We find that layers 3 and 4 have a much lower MSE and a much larger cosine similarity compared to other layers in the model. Hence, these layers somehow 'store information' about the masked input matrix.

We also probe for the true matrix element at missing entries, and find that the hidden states at these positions get gradually more correlated with depth (through linear probing). Further, we also attempt to extract information about singular vectors of the ground truth matrix from the hidden states through linear probing, though are unable to conclusively do so. We discuss these results in Appendix K.

### 3.3 Role of Embeddings

**Token Embeddings** The $\ell_2$ norm of token embeddings corresponding to values from $-1.5$ to $1.5$ is symmetric w.r.t. $0$ as seen in Fig. 7a. Further, the PCA of token embeddings in Fig. 7b shows that the embeddings have a separable structure based on the sign of the real–valued input (y–axis), and continuous variation w.r.t. magnitude of input (x–axis). Importantly, unlike other metrics, token embeddings do not seem to abruptly change only at step $15000$; rather, the final structure appears before the sudden drop in loss. Similar to [27], we compute the top–2 principal components of the token embeddings at the final step ($50000$), and project the token embeddings at intermediate training

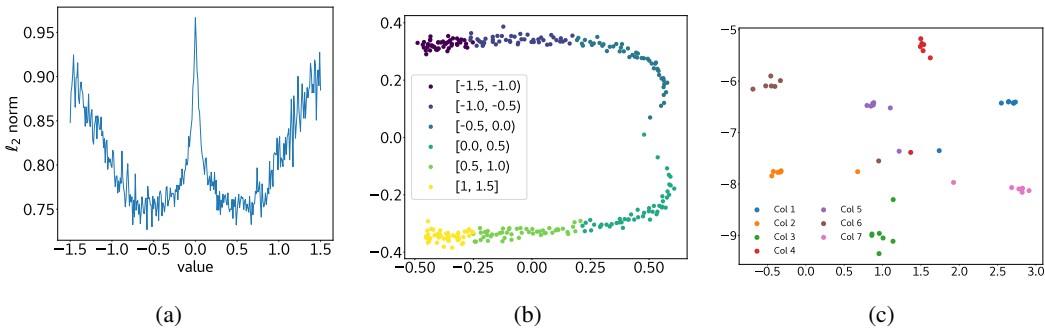

(a)          (b)          (c)

Figure 7: **Embeddings demonstrate relevant structure.** In the post–shift model, positional and token embeddings exhibit properties demonstrating that the model has learnt relevant information about the matrix completion problem. (a) $\ell_2$ norm of token embeddings is symmetric around $0$. This aligns with the intuition that the norm of token embeddings should depend only on the magnitude of the input, and not on its sign. (b) Top–2 principal components of token embeddings correspond to the magnitude and sign of the real valued input. In our case, the 'y-axis' denotes sign of input, and the 'x-axis' denotes the magnitude of the input value. (c) Positional embeddings of elements in the same column cluster together in the t–SNE projection, showing that the model uses positional information relevant to the matrix completion problem.

steps on these components. The results (Fig. 9, Appendix C) show that the embeddings align very closely to the final arrangement before the actual drop. This is as expected, since the model needs to learn what the tokens actually represent on the real line, before it can use those values for completing missing entries. This also explains to some extent why the model implements copying before the sudden drop, since accurately learning token embedding-unembedding is sufficient for that task.

**Positional Embeddings**    In the t-SNE projection of positional embeddings, positions in the same column tend to cluster together as seen in Fig. 7c. This is non–trivial because we have not used any marker tokens to mark the end of a row or column. Further, note that in contrast to token embeddings, positional embeddings do not have a continual evolution in structure – Fig. 10 (Appendix C) shows that the clustering appears only after the sudden drop (step 20000 and after). This along with the evolution of attention heads (Sec. 3.1, 3.2) aligns with how the pre–shift model copies observed entries with little effect from ablating attention heads or positional embeddings (Sec. 3.2.1).

# 4   Sudden Drop in Loss – Role of Model Components

Is it possible to analyse training dynamics of individual model components to derive insights about the full model training? This is motivated by the findings in the previous section on embeddings, and in Section 3.1; the pre–shift model does not use Attention layers for its computation in that stage, and relies on other components to copy input entries. Hence, in our case, the sudden drop corresponds in large part to learning the right Attention patterns (see Appendix M.1). To analyse training dynamics of different model components, we choose (a set of) components – Attention layers, MLP layers, Positional Embeddings and Token Embeddings, randomly initialize them and freeze the weights of other components to their values at the final step of training (Fig. 8).

We find that (a) MLPs and Token Embeddings converge without any observable plateau or sudden drop in loss; (b) for other components, the dynamics resemble those for the full model training (i.e. plateau and then sudden drop), and (c) Positional embeddings show the longest plateau in loss.

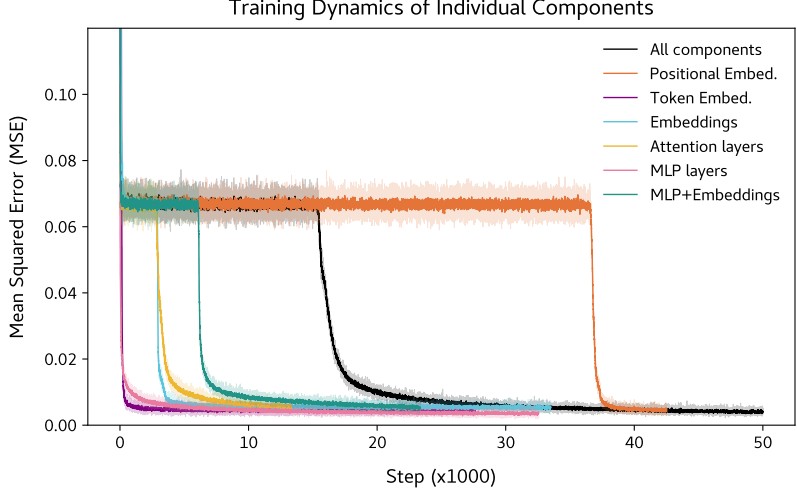

Figure 8: **Individual model components have distinct training dynamics**. Training individual model components, initializing others to their final value ('All components' indicates normal training). There is no loss plateau for token embeddings and MLP layers, in contrast to positional embeddings, where the sudden drop occurs just before step 40000. In all other cases the sudden drop occurs before the sudden drop in usual training.

**Additional Results**    To further understand the effect of data and model properties on the sudden drop in loss, we train

- a 2–layer, 2–head GPT model on the matrix completion task (Appendix G);
- models of different depth (number of layers) and width (hidden state dimension) (App. H);

- our model on (mixture of) matrices of different sizes keeping the rank fixed at 1 (App. I);

- a 12-layer, 12-head model on $10 \times 10$ matrices of multiple ranks (separately) (App. I);

- our model on input matrix entries $(U_{ij}, V_{ij})$ i.i.d. $\sim \mathcal{N}(0, 1)$ instead of Unif$[-1, 1]$ (App. J.1); we also analyze test–time OOD performance of our model in App. J.2.

## 5  Related Work

[27, 28, 32, 41, 38] analyse 'grokking', the sudden emergence of generalization during model training. In the context of training dynamics of MLM, [8] analyses 'breakthroughs' (sudden drop in loss and associated improvement in generalization capabilities of the model), specifically for BERT. They show that the breakthrough marks the transition of the model to a generalizing one. Their work however is focused on language tasks, distinct from our setting which is mathematical (and hence more controllable) in nature. We also note that their work is not in the online training setting; our setup is online in the sense of sampling new data at every step of training.

Mathematical problem solving capabilities of Transformers have been a topic of interest lately [24, 7, 4]. In fact, [24] show that learning addition from samples is equivalent to low–rank matrix completion. Further, [7] show that it is possible to train a transformer based model to solve various linear algebraic tasks e.g. eigendecomposition, matrix inversion, etc.; however, to the best of our knowledge, interpretability studies for such tasks have not been conducted before. For interpretability in simpler math tasks, [18] mechanistically analyse GPT-2 small on predicting whether a number is 'greater-than' a given number, by formulating the problem as a natural language task. [35, 36, 10] analyse BERT from an interpretability perspective. More recently, there has been a line of research works analysing decoder based models to reverse–engineer the mechanisms employed by these models, termed as 'mechanistic interpretability' [14, 31, 32, 39, 11, 33, 25, 26, 23, 34, 20]. We note that our setting is distinct from the recent work on solving mathematical tasks like linear regression through 'in–context' learning in transformers [4, 1, 9, 15, 17, 2, 30, 37]. Whether our model learns to implicitly 'implement' an optimization procedure as shown in some of these is an open question. We discuss related work in more detail in Appendix E.

## 6  Conclusion

We trained a BERT model on matrix completion, and analyzed it before and after the sudden drop in training loss (algorithmic shift) to interpret the algorithm being learnt by the model, and gain insight on why such a sharp drop in loss occurs.

It is evident in our analysis that both before and after the shift, the model does not really compute anything at observed positions, and simply copies these entries. For missing entries, we have shown that the model learns useful abstractions rapidly through the algorithmic shift. Mathematically formulating what algorithm the model employs to implement matrix completion for missing entries is a direction for future work.

Since our work is primarily interpreting model training and mechanism, all experiments are with small scale matrices (largest being $15 \times 15$), and the current method would likely need modifications to scale to larger matrices. Finally, we only intended to study Transformers on matrix completion as a toy task from an interpretability viewpoint, and do not advocate replacing existing efficient solvers for matrix completion with our approach.

**Societal Impact**   We study Transformer based models on their ability to solve a mathematical task (matrix completion) and the associated training dynamics. The work focuses on Transformer interpretability, aiding in improving our understanding of these models and their training and thus we do not foresee any negative societal impact of our work.

**Acknowledgements**   We thank Yu Bai, Andrew Lee, Naomi Saphra and anonymous reviewers for their helpful comments. WH acknowledges support from the Google Research Scholar Program. ESL's time at University of Michigan was supported by NSF under award CNS-2211509 and at CBS, Harvard by the CBS-NTT Physics of Intelligence program.

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

# A Copying in Pre–Shift Model

| Step | Input Samples | Mask = "MASK" | | Mask = "0.44" | | Mask = "–0.24" | |
|---|---|---|---|---|---|---|---|
| | | $L'_{mask}$ | $L_{obs}$ | $L'_{mask}$ | $L_{obs}$ | $L'_{mask}$ | $L_{obs}$ |
| 1000 | Rank−2 matrices | 1.4e-3 | 9.3e-4 | 7.6e-4 | 1e-3 | 6.7e-4 | 9.6e-4 |
| | Random matrices | 1.5e-3 | 8.3e-4 | 7.8e-4 | 1e-3 | 6.8e-4 | 9.6e-4 |
| 4000 | Rank−2 matrices | 3e-4 | 3.3e-4 | 4e-4 | 2.8e-4 | 3.7e-4 | 2.7e-4 |
| | Random matrices | 2.8e-4 | 3.5e-4 | 3.7e-4 | 3e-4 | 3.6e-4 | 2.8e-4 |
| 14000 | Rank−2 matrices | 1.6e-5 | 3.4e-5 | 1.8e-5 | 4.9e-5 | 5.1e-6 | 6.7e-5 |
| | Random matrices | 1.1e-5 | 3.7e-5 | 3.0e-5 | 8.5e-5 | 4.1e-6 | 1.1e-4 |

Table 1: Models at different steps before sudden drop implement copying, predicting the value for mask token at missing entries.

# B Nuclear Norm Minimization

We use the regularized version of the nuclear norm minimization problem as detailed in Sec. 3.2, and obtain the following $L, L_{obs}, L_{mask}$ for various values of $\lambda$. We average our results over 256 samples generated in the same way as the training data for BERT (including rounding off to 2 decimal places) for the sake of comparison.

| $\lambda$ | $L_{obs}$ | $L_{mask}$ | $L$ |
|---|---|---|---|
| 0.0005 | 1.015e−5 | 0.040728 | 0.012173 |
| 0.001 | 3.686e−5 | 0.040456 | 0.01211 |
| 0.0015 | 7.959e−5 | 0.040505 | 0.012155 |
| 0.002 | 0.00013769 | 0.040734 | 0.012264 |
| 0.005 | 0.00078591 | 0.043402 | 0.013516 |

# C Embeddings Progress during Training

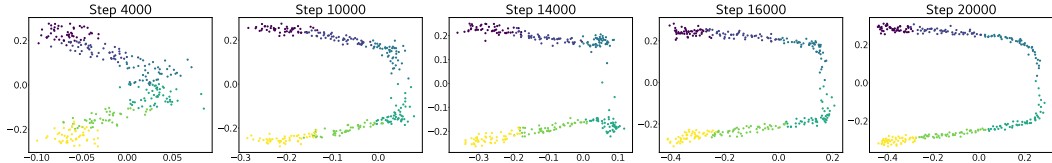

Figure 9: **Token embedding structure appears before sudden drop.** Projection of token embeddings along principal components of embeddings at step 50000. (Labels same as Fig. 7). Embeddings align with the principal components early on in training before the sudden drop in loss.

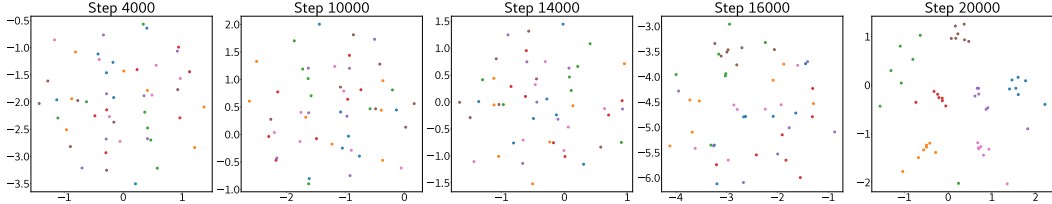

Figure 10: **Positional embedding structure appears after sudden drop.** Evolution of t-SNE projection of positional embeddings with training. (Labels follow the same color coding as Fig. 7). In this case, the positional embeddings show clustering some time after the sudden drop in loss has occurred at step 15000.

# D   Experimental Details

**Online Training**   In online training, the data is sampled afresh from the distribution at every step. Since data is not partitioned into fixed train and test sets, we only analyze the training loss in all cases.

**Tokenizing Matrices**   For tokenizing real values, we discretize the range $[-10, 10]$ in steps of size $\epsilon = 0.01$, and assign token IDs starting from 1; the mask token (MASK) is assigned ID 0. Input to the model is the tokenized masked sequence $X_{mask} = \text{TOK}\left(\text{Vec}(X \odot M)\right)$, where $\odot$ denotes the element-wise product, Vec denotes vectorizing the $n \times n$ matrix to a $n^2$-dimensional vector, and TOK denotes tokenization. Due to this preprocessing, in all cases $X_{ij}$ is rounded to 2 decimals for computing $L$.

# E   Related Work

In the online training setup, [5] study the parity learning problem using a variety of model architectures, and show that 'hidden progress measures' can be used to track abrupt changes in model performance during training. [29] study abrupt learning in an autoregressive (GPT), language data setup, connecting learning the grammar to percolation on graphs. [19] discuss abrupt learning dynamics in the context of transformers and claim that the softmax function in Attention leads to longer training loss plateaus – however, reducing the length of plateau does not explain why the drop in loss is sudden and sharp when it occurs. [43] show that initialization of the model affects whether it learns to infer the compositional structure of the task, or simply memorizes the solution.

[4] show that in an in–context learning framework, a transformer based model can learn to select the most optimal statistical method for solving the task in the prompt, without explicitly being provided any information about the optimal method (called 'in–context algorithm selection' in their work). We emphasize that our setup is not in–context learning, and is quite distinct from [4] as far as the task being solved is concerned. However, whether the framework of layer-wise in-context gradient descent can also be used in our setup is a plausible and open direction for future work.

In [7], the author shows empirically that an encoder-decoder transformer can be trained to solve various linear algebraic tasks, such as eigendecomposition, SVD, matrix inverse etc. They support their findings by showing that the model generalizes to matrix distributions outside the training distribution to some extent, and that OOD performance can be improved by training on non-Wigner matrices. While many experiments in [7] also show a sudden jump in accuracy (Fig 1,2), they do not analyze why such a sudden jump occurs during optimization. In our work, we analyze the sudden drop and the model before and after it to derive insights into the sudden drop in loss in our setup.

[24] show that even a small transformer model can be trained to perform arithmetic operations like add, subtract, multiply accurately through appropriate data selection and formatting, and using Chain-of-Thought prompting. They further show that learning addition is connected to rank-2 matrix completion, and that the sudden jump in accuracy with increasing number of observed entries of the matrix is recovered when their model is trained on datasets of different sizes. This is because the size of the dataset for addition can be seen as the number of observed entries of the rank-2 matrix representing the addition table. We point out that while the task in this case is related to matrix completion, ours is a completely different setup, where the sudden drop happens with the number of training steps with each step consisting of 256 low-rank matrices, each with a fixed fraction $(p_{\text{mask}})$ of observed entries.

# F   Causal Intervention on Attention heads

In the uniform ablation setup, it is possible that setting the softmax probabilities to a given value might change the distribution of resultant hidden states, and consequently degrade model performance. A more principled technique to analyze the effect of a specific component is to replace the hidden state just after that component by hidden states on a different input, and analyze how this affects the final output [42]. In our case, we intervene on attention heads by replacing the hidden state after an attention head for input matrix $X$ by the hidden state for input $(-X)$. Importantly, this change does not affect properties like rank of the input, and hence the hidden states obtained are from the same distribution as those for input $X$.

Step 1  Extract the hidden states for all attention layers from the model on some input matrix $X$; call these $h_+$. Concretely, these hidden states are obtained just after the matrix product of the softmax attention probabilties and the value matrix and hence before the output matrix product.

Step 2  Change the input to the model to $-X$, however, also replace the hidden states *just after* the attention layers with $h_+$ obtained in Step 1. Call the output of the model in this setup as $f_p(-X, X)$.

We observe that, the MSE between $f_p(-X, X)$ and $X$, averaged over 256 samples at masked positions is approximately $0.014$ (this is comparable to optimal $L_{mask}$), compared to the MSE between $f_p(-X, X)$ and $-X$ being $0.8066$. This demonstrates that the attention heads are causally relevant to the model output for missing entries.

## G   Autoregressive Setup

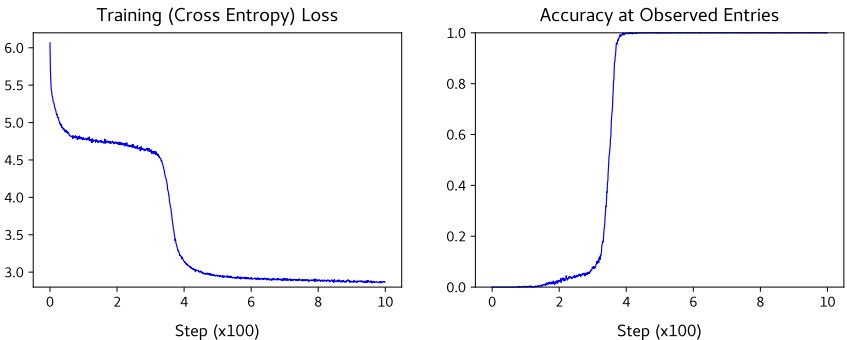

Figure 11: **GPT also shows abrupt learning for matrix completion**. Training a 2–layer, 2–head GPT model on $7 \times 7$, rank$-2$ matrices in the autoregressive training setup. Here, the model is trained using cross–entropy loss in a next–token prediction setup over full input sequences of the form $\tilde{X}_{11}, \tilde{X}_{12}, \ldots, \tilde{X}_{77}, [SEP], X_{11}, X_{12}, \ldots, X_{77}$ where $\tilde{X}, X$ are partially observed and fully observed matrices, flattened and tokenized as in the BERT experiments. We find that the sudden drop corresponds to the model learning to copy the observed entries in the input matrix. While we could not achieve performance comparable to BERT for missing entries, we believe it should be possible with some modifications to the training setup.

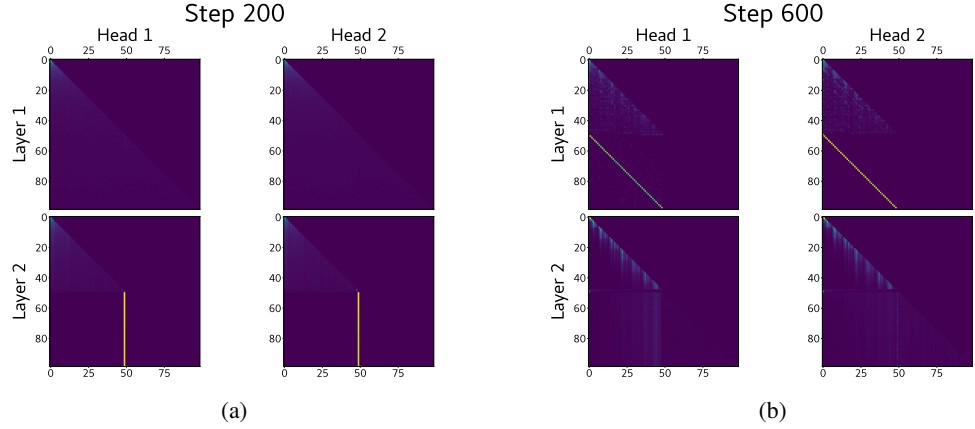

Figure 12: **Attention heads demonstrate sudden change in matrix completion using GPT.** We find that even in the GPT case, the attention heads change from trivial (Layer 2, attending to the [SEP] token for all positions in the output) to those in Layer 1 attending to the corresponding positions in the input ($\sim$identity maps). This corroborates with our finding about the model learning to copy observed entries after the sudden drop, in Fig. 11.

## H  Effect of Model Size

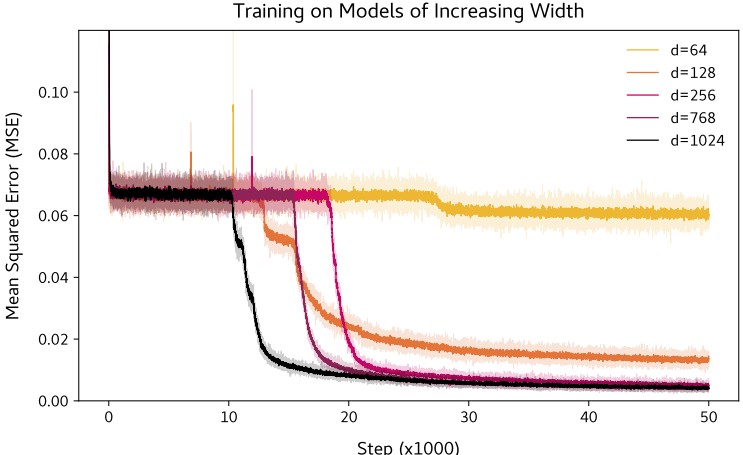

Figure 13: **Effect of model width.** Training with different model widths (hidden state dimensionality) on $7 \times 7$ rank$-2$ inputs. The plot demonstrate that $d = 64$ is too small to obtain accurate matrix completion, and that the performance is sub–optimal for $d = 128$. We scale the hidden layer width of the $2-$layer MLPs as $4d$, as is done in practice.

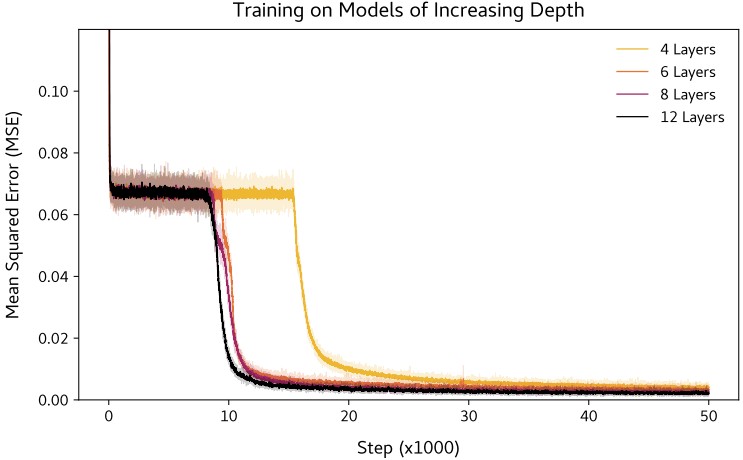

Figure 14: **Effect of model depth.** Training with different model depths (number of layers) on $7 \times 7$ rank$-2$ inputs. The plot demonstrate that as depth increases from $4$ to $6$, the sudden drop occurs earlier, but increasing depth beyond this $(8, 12)$ has little effect. The final MSE obtained also follows the intuitive ordering (largest for $L = 4$ decreasing with $L$ upto $L = 12$; though the variation is not significant.

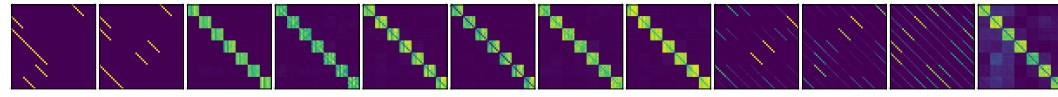

Figure 15: **Using 1 attention head per layer.** We find that training a $12-$layer, $1-$head BERT model on matrix completion leads to similar loss $(4e-3)$ and attention heads as the $4-$layer, $8-$head model.

# I  Effect of Matrix Size and Rank on Training

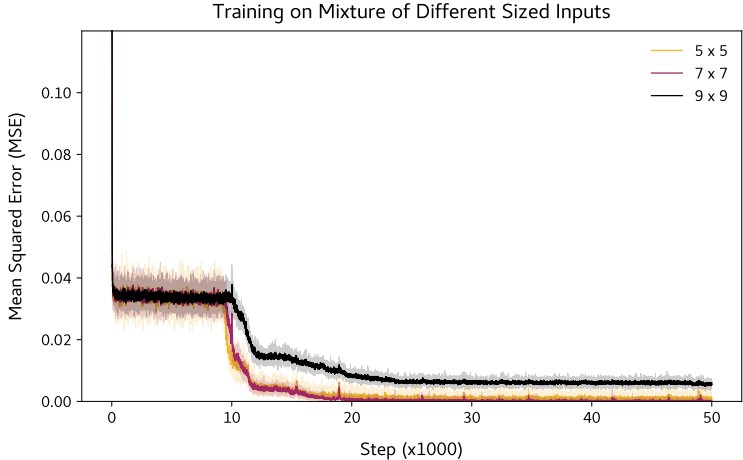

Figure 16: **Learning order when trained on mixture of matrices.** Training on uniform mixture of $5 \times 5, 7 \times 7$ and $9 \times 9$ rank$-1$ matrices i.e., at each step, 256 samples of size $n \times n$, with $n$ chosen randomly from $\{5, 7, 9\}$. The plots show the test set MSE on separate 256 samples of the 3 different matrix sizes.

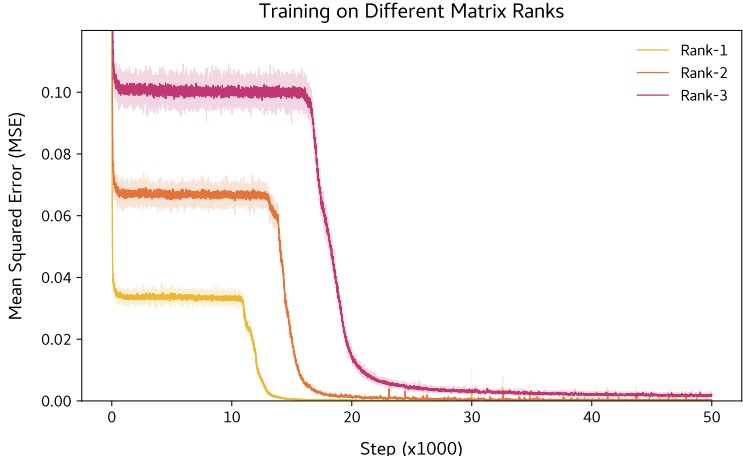

Figure 17: **Effect of problem complexity**. Training a $12-$layer, $12-$head model on $10 \times 10$ matrices of rank $r = 1, 2, 3$. There is a clear progression in terms of the training step where sudden drop occurs, and the final loss values scale roughly as $L \sim c \cdot 10^r$, $c \approx 2 \times 10^{-6}$. This also predicts that $L \sim 0.02$ for $r = 4$, i.e. the model does not solve matrix completion to low MSE, which is what we also observe in practice.

## J  Effect of Input Distribution

### J.1  Training

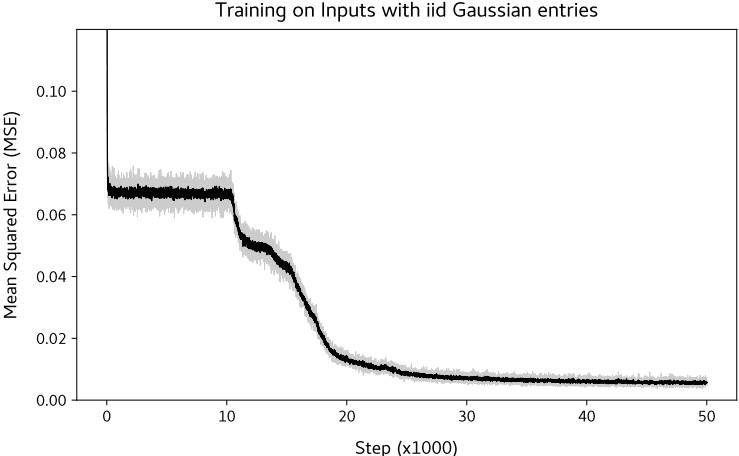

Figure 18: **Sudden drop is not limited to uniform distribution**. Training on i.i.d. $\mathcal{N}(0, 1)$ entries. We find that the sudden drop also occurs in this case, and the final loss value $\sim 5.6 \times 10^{-3}$, similar to the value obtained for i.i.d. Uniform$[-1, 1]$ entries.

### J.2  Inference

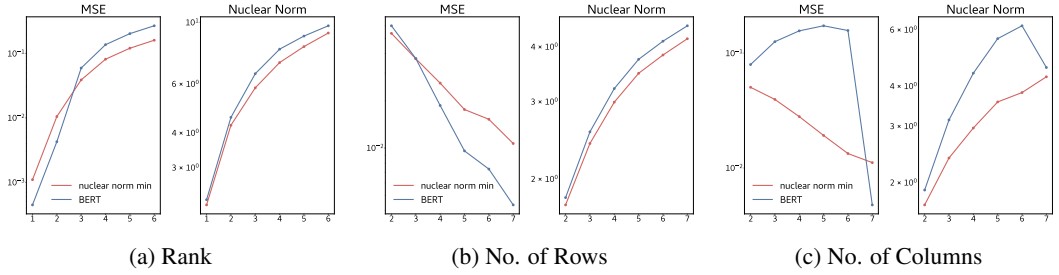

    (a) Rank                  (b) No. of Rows              (c) No. of Columns

Figure 19: **Model performs similar to nuclear norm minimization on OOD samples.** OOD performance at inference for various values of rank, number of rows and columns of the input matrix. Except (c), the OOD performance of the model is close to the nuclear norm minimization solution for the same inputs. For (c), since we observed that positional embeddings depend on the column of the element, changing the number of columns adversely affects performance.

**Matrix Distribution**  We also change the input distribution of the matrix entries to Gaussian and Laplace, and measure average MSE over $1024$ samples of size $7 \times 7$ and rank$-2$, to evaluate the OOD performance of the trained model. We find that

- for entries i.i.d. $\sim \mathcal{N}(0, 0.25)$, $L \approx 4 \times 10^{-3}$, and
- for entries i.i.d. $\sim \mathrm{Laplace}(0, 0.25)$, (parameterized by mean and scale), $L \approx 2 \times 10^{-3}$.

That is, the OOD performance on these distributions is similar to the MSE obtained for the in–training distribution (Uniform$[-1, 1]$).

# K   Probing: Additional Results

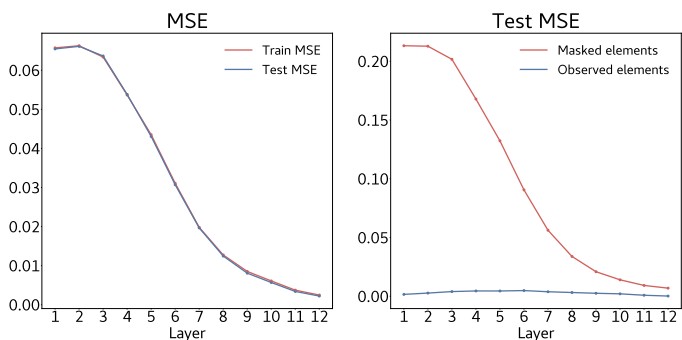

Figure 20: **Hidden states encode the true value at missing entries.** Probing for the corresponding element in the fully observed matrix $X$. (Left) comparing the train and test MSE of the linear probe, to confirm that the probe does not overfit. (Right) Test MSE evaluated on the masked elements, that shows an interesting variation: the MSE goes down at a nonlinear rate, hinting that implicit layer–wise optimization could be taking place.

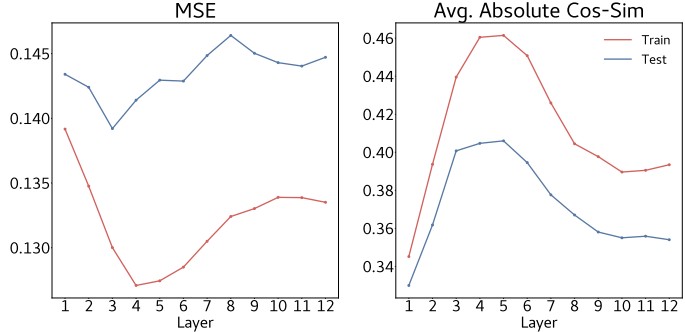

Figure 21: **Model does not encode singular vectors in hidden states**. Probing for the first left singular vector at all positions; that is, for fully observed input matrix $X \in \mathbb{R}^{n \times n}$ with SVD $X = U\Sigma V^\top$, we probe for $u = [U_{11}, U_{21}, \ldots, U_{n1}]^\top \in \mathbb{R}^n$. (Left) Train and Test MSE across different layers of the model; (Right) Average Absolute Cosine similarity of the predicted $u$ with the actual $u$. Both evaluations are inconclusive, since the MSE is too large, and the cosine similarity is not much larger than the average absolute cosine similarity ($\approx 0.3$) between 2 i.i.d. vectors $\sim \mathcal{N}(0, I_{7 \times 7}) \in \mathbb{R}^7$

# L   Ablating Groups of Structured Attention Heads

| Group of Attention Heads Ablated | L | | $L_{obs}$ | | $L_{mask}$ | | Ratio of $L$ (w/ to w/o) |
|---|---|---|---|---|---|---|---|
| | w/ ablation | w/o ablation | w/ ablation | w/o ablation | w/ ablation | w/o ablation | |
| (2,1), (3,4), (4,8) | 0.0073 | 0.0035 | 0.0001 | 9e−5 | 0.0245 | 0.0117 | 2.09 |
| (4,3), (4,4) | 0.0079 | 0.0045 | 3e−4 | 8.8e−5 | 0.026 | 0.015 | 1.76 |
| (2,2–4), (2,6), (3,2), (3,3), (3,5), (3,1), (3,6) | 0.057 | 0.0043 | 2.1e−4 | 9e−5 | 0.1885 | 0.0142 | 13.26 |
| (2,5), (2,7), (2,8) | 0.0112 | 0.0049 | 8.5e−5 | 9.2e−5 | 0.037 | 0.016 | 2.29 |
| (1,1–2), (1,5–8) | 0.0314 | 0.0048 | 1.7e−4 | 8.8e−5 | 0.1038 | 0.0157 | 6.54 |

# M   Attention Heads

## M.1   Variation along training

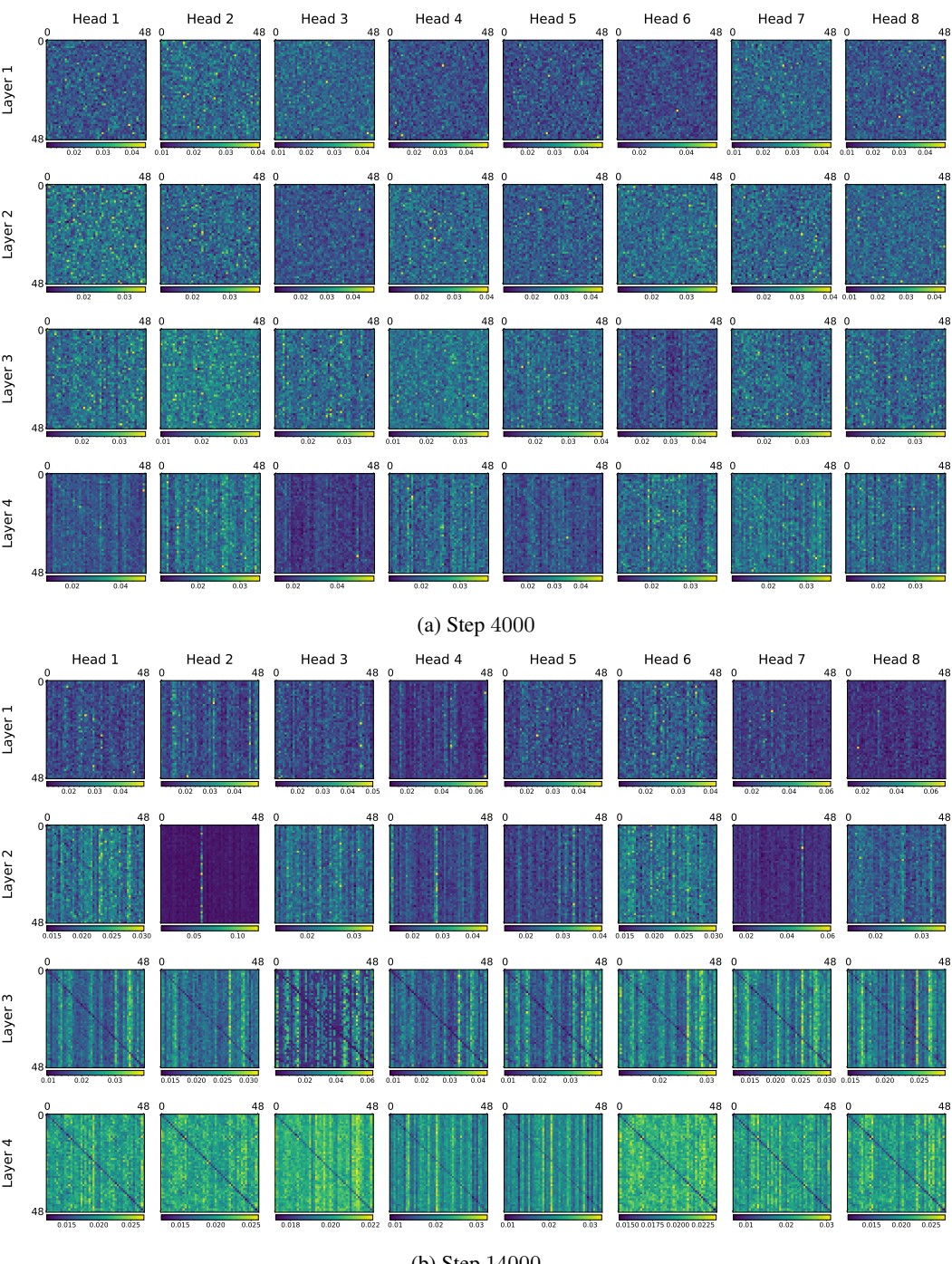

(a) Step 4000

(b) Step 14000

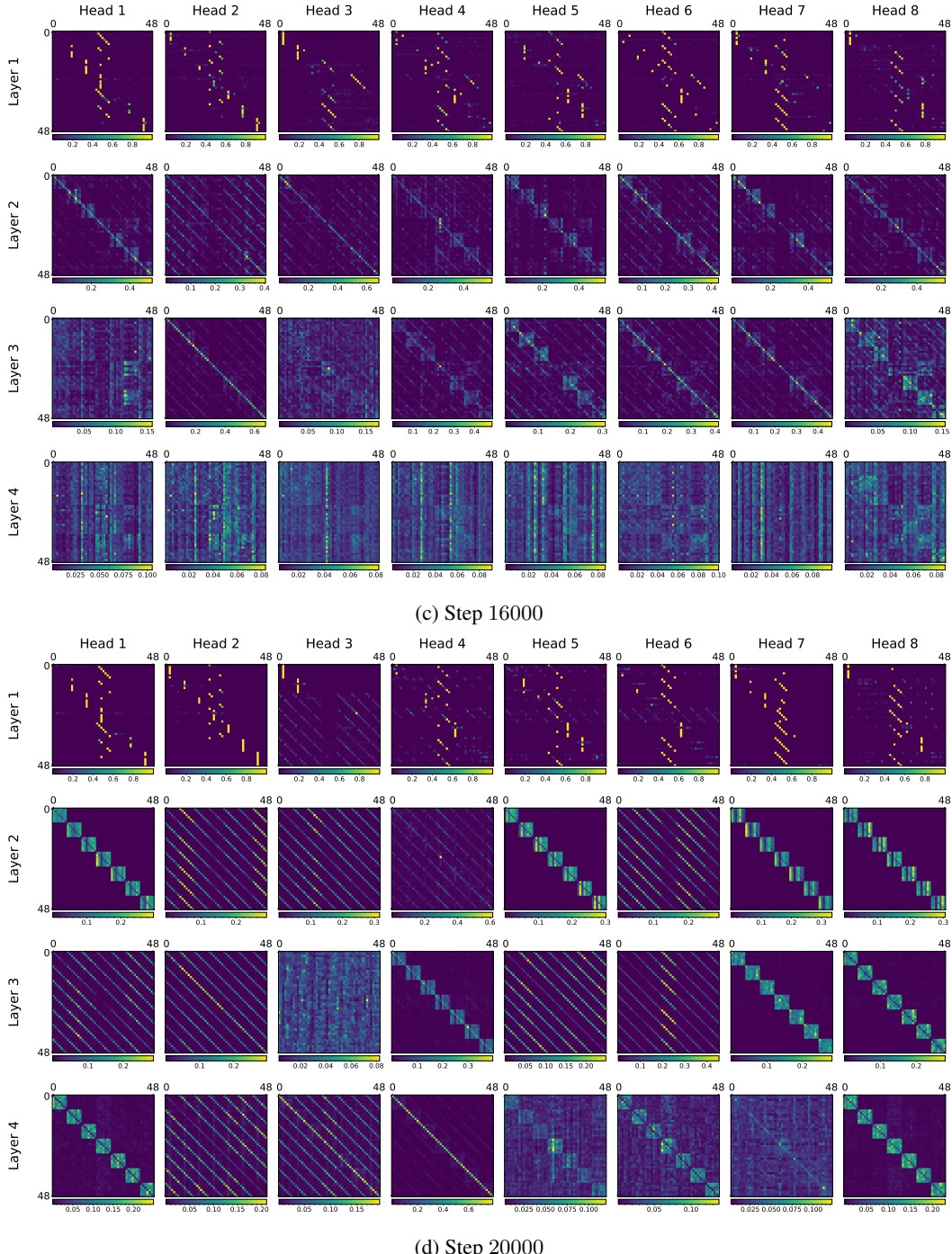

(c) Step 16000

(d) Step 20000

Figure 22: Attention heads across various training steps.

## M.2   Effect of changing mask structure

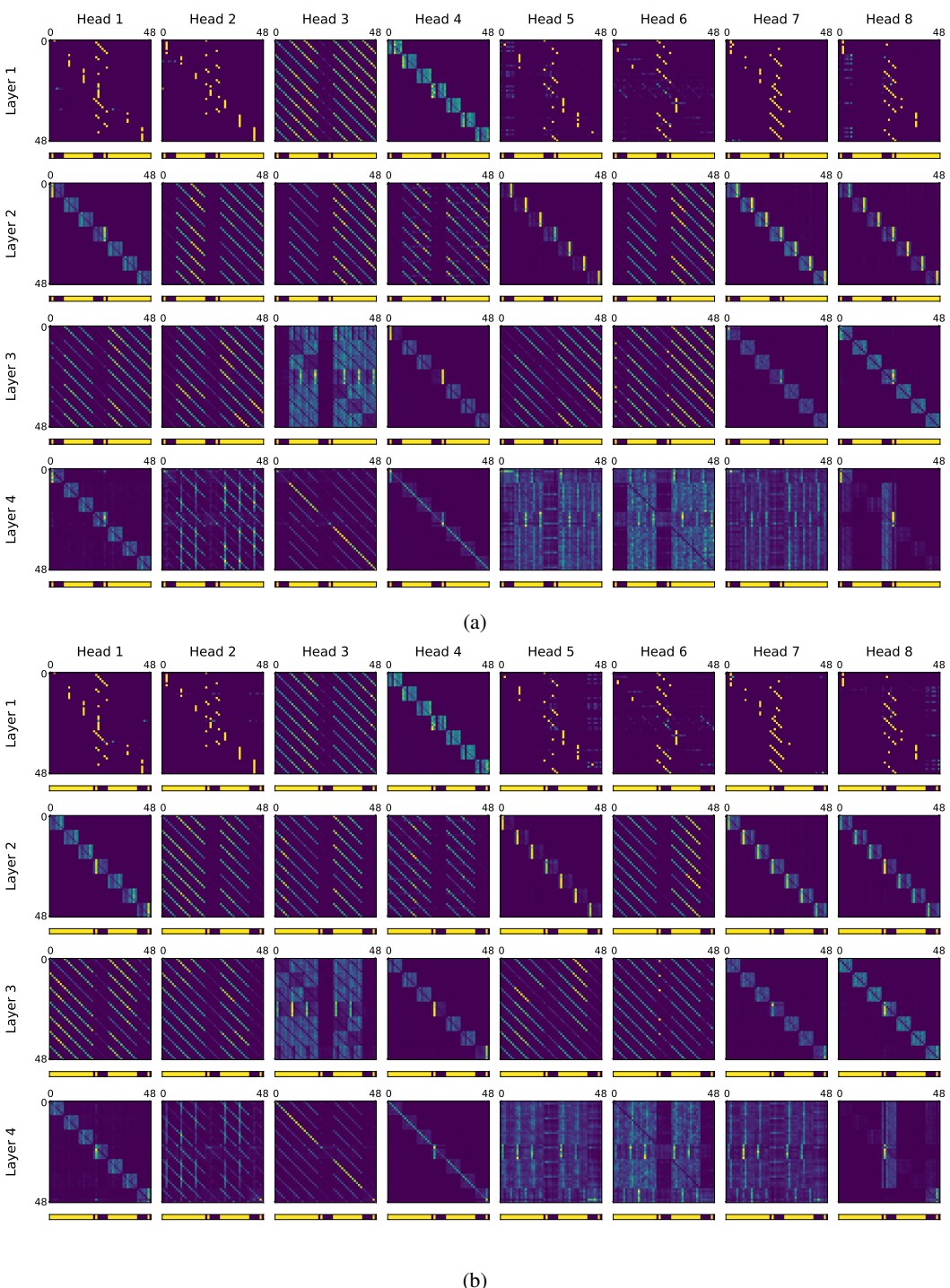

(a)

(b)

Figure 23: Attention heads and corresponding masks; blue denotes masked position in the input matrix.

# N  Attention Heads for larger inputs

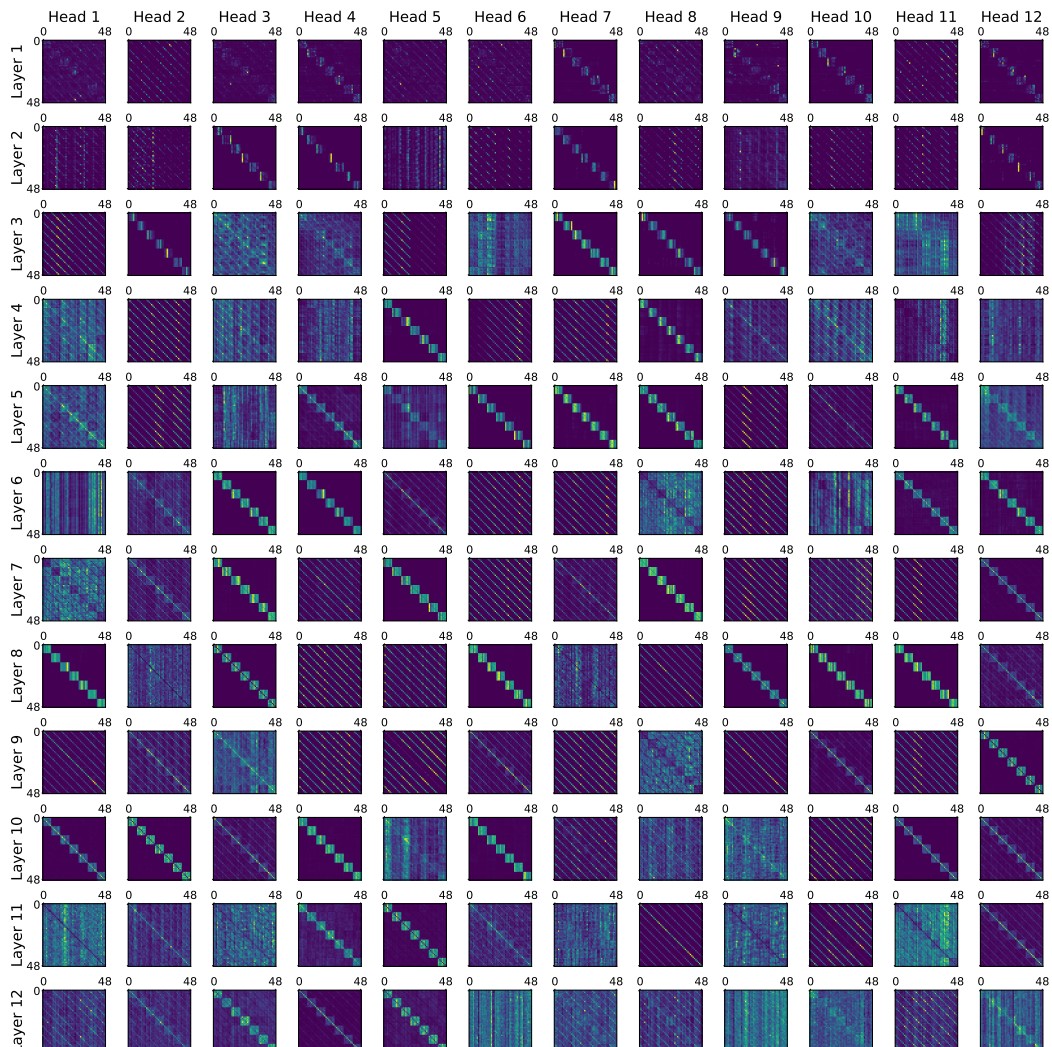

Figure 24: Attention heads in 12 layers, 12–heads model on $7 \times 7$ rank–2 input

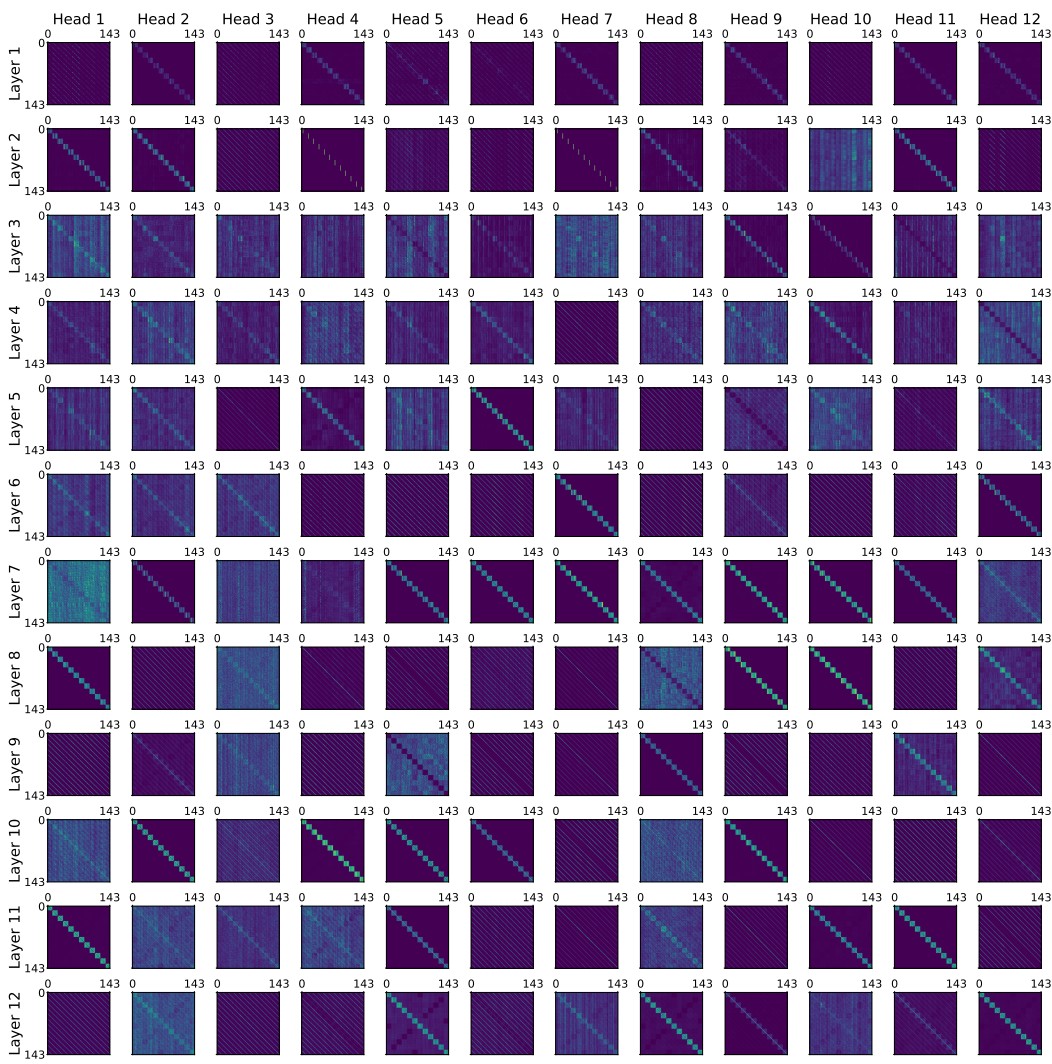

Figure 25: Attention heads in 12 layers, 12–heads model on $12 \times 12$ rank–3 input

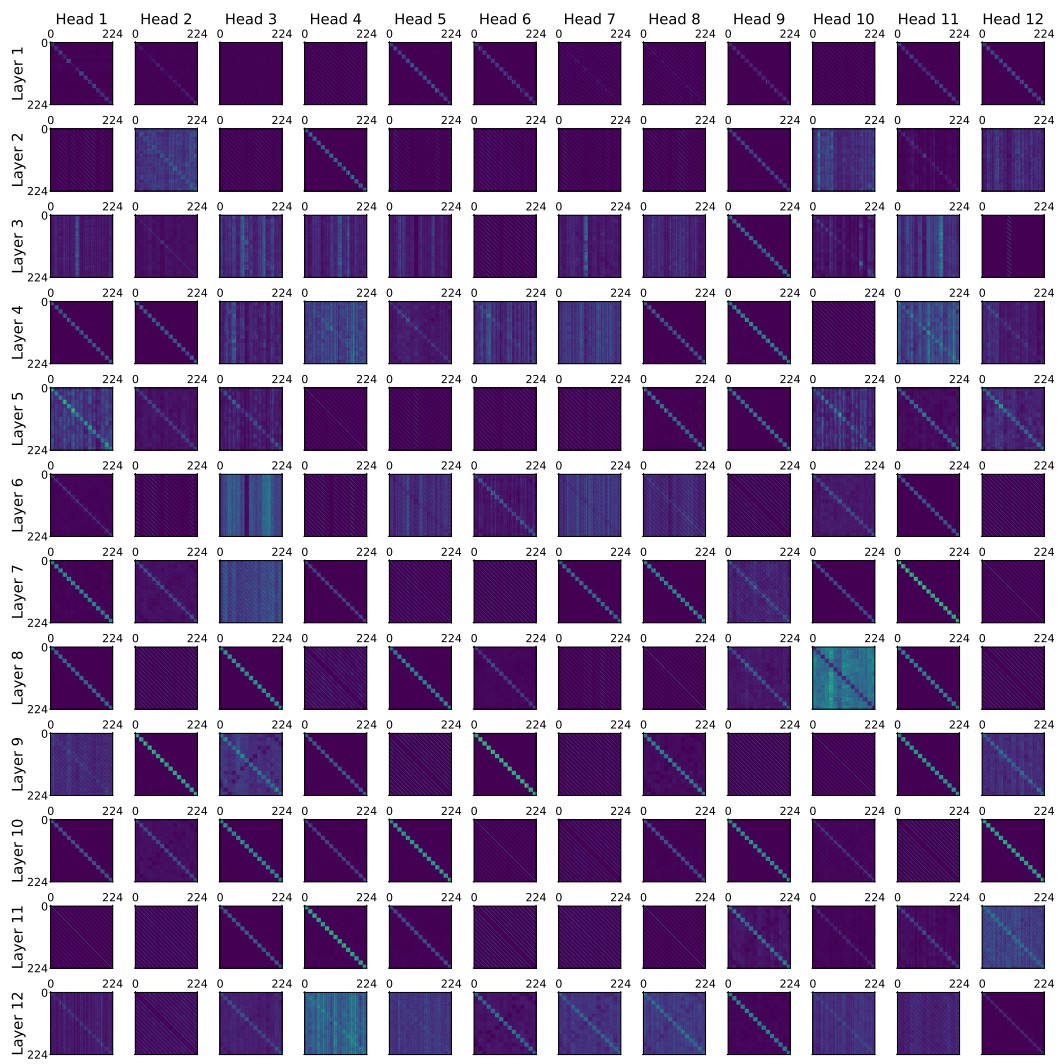

Figure 26: Attention heads in 12 layers, 12–heads model on $15 \times 15$ rank–4 input

