# OpenReview forum: "Abrupt Learning in Transformers: A Case Study on Matrix Completion"
_NeurIPS.cc/2024/Conference — NeurIPS 2024 poster_

### Official Review · Reviewer_eRgv · 2024-06-27

**Soundness:** 3
**Presentation:** 3
**Contribution:** 2
**Rating:** 5
**Confidence:** 3

**Summary:**

The paper explores the behavior of Transformer models in the context of low-rank matrix completion, treated as a masked language modeling (MLM) task. The authors train a BERT model to solve matrix completion and analyze its performance, particularly noting an algorithmic shift characterized by a sudden drop in loss, transitioning from copying input to accurately predicting masked entries. The paper aims to provide insights into the interpretability of Transformer models.

**Strengths:**

1. The paper presents an interesting approach by framing matrix completion as an MLM task.

2. The experiments are detailed, with clear descriptions of data preprocessing, training, and performance metrics.

3. The identification of an algorithmic shift during training is a significant observation, providing insights into the model's learning dynamics.

**Weaknesses:**

1. The paper lacks a deep theoretical or mechanical analysis of why the model undergoes an algorithmic shift and how it learns the task.

2. There is insufficient explanation of the internal mechanisms by which the model learns matrix decomposition, particularly in the second stage.

3. The experiments are conducted on relatively small-scale matrices (up to 15x15), raising concerns about the scalability and generalizability of the findings.

**Questions:**

1. Does the model operate through two distinct pathways for non-MASK and MASK positions, as suggested by the different behaviors observed during training?

2. What would be the model's behavior if the attention matrices' parameters were frozen at the start of training to match those of a fully trained model? Would the model still exhibit the initial copying behavior, or would it directly perform matrix completion?

3. How well does the model generalize to different types of low-rank matrices, especially matrices with different sparsity and rank levels? Can it accomplish the task of finding the lowest rank matrix?

4. Can the approach be scaled to larger matrices, and what modifications would be necessary to achieve this?

**Limitations:**

See weaknesses.

---

> ### Author Rebuttal · Authors · 2024-08-07
>
> We thank the reviewer for their comments, and are excited to know that they found our approach interesting!
>
> - *The paper lacks a deep theoretical ...*
>
> We would like to point out that apart from the analysis in our paper, to the best of our knowledge there has been no work on analyzing such an algorithmic shift either mechanistically or theoretically. Ours is the first step in this direction where we start by formulating a mathematical setup, since this gives us a clearer picture of mechanisms learnt by the model before and after the algorithmic shift. We provide several detailed experiments on this front, as seen in Sections 4–6 of the paper.
>
> To address reviewer concerns, we have also added 2 additional probing experiments that analyze the hidden states in the model in more detail than our existing probing experiment. To understand why the model undergoes algorithmic shift, we have now additionally added interventional experiments wherein various combinations of model components are frozen to their final parameter states (i.e., from the final step of training). These results further indicate rapid structural changes in attention layers drive sudden loss drops. We have also added several other experiments: e.g., OOD generalization experiments, training on different matrix distributions, scaling model width / depth. All these results further demonstrate the generalizability and robustness of our results. Please see our global response for more details on these experiments.
>
> - *The experiments are conducted ...*
>
> Please see our global response on “Larger matrices’, where we discuss possible ways to scale up our approach to larger input matrices.
>
> - *Does the model operate …*
>
> Our results show that this is most likely the case, since the mechanisms for non-MASK and MASK prediction remain distinct after the model has converged (copying observed entries vs. computation). We demonstrate this empirically through intervention experiments on the model weights in Sec 5.1 for observed (non-MASK) entries, and in Sec 5.2 for MASK entries. One of the main takeaways is that prediction at observed entries is not affected significantly by attention layers (since uniform ablating them does not lead to a drastic change in performance), whereas prediction at masked entries is significantly affected. Please see Sec 5 for more details about these experiments.
>
> - *What would be the model's behavior …*
>
> Please refer to Fig. K in the attached PDF; we find that freezing only the attention layer to weights obtained at the last training step and training only MLPs and embeddings (labeled 'mlp+embed’ in the plot) still leads to a plateau and then sudden drop. However, on freezing both Attention and Embedding layers, and training only MLP layers (labeled 'mlp’), we do not see plateau or sudden drop in loss, and the training loss decreases in a continuous manner to optimal value.
>
> Our hypothesis is that the sudden-ness of the drop is contributed to by the Attention / Embedding layers, due to these components encoding ‘structure’ of the matrix completion problem. To corroborate this hypothesis, we also do the following experiments. Below, training all components is labeled `all’, which is the loss plot in the submitted manuscript – it is a reference for visualizing “sudden-ness” of drop in loss, and the length of loss plateau.
>
> **Train only Attention layers (labeled ‘att’)** – smaller loss plateau than “all” with sharp drop
>
> **Train only Embeddings (both positional and token, labeled ‘embed’ in the plot)** -  smaller loss plateau than “all” with sharp drop
>
> **Train only positional embeddings (labeled ‘pos’)** – this has the longest plateau of all settings
>
> **Train only token embeddings (labeled ‘token’)** – there is no observable plateau or sudden drop in loss. Please note that this also matches the observation in our submission (Section 6, “Do embeddings change abruptly?”) where we have plotted the projection of token embeddings on the principal components of the final model’s token embeddings. We observed that the structure of this 2D projection does not change abruptly at step 15000, and that is further supported by this modified training process, which shows that token embeddings are learnt by the model without any significant observable plateau or sudden drop. Whether this early learning of token embeddings is also a driving force behind the sudden drop, is still an open question in our setup.
>
> From these observations, a mechanistic hypothesis emerges: positional embedding layers and attention layers are the main contributors towards the sudden drop dynamics of the training loss. Formalizing and rigorously verifying these observations is an important direction for future research.
>
>
> - *How well does the model generalize to different types …*
>
> We find that the model performs well on test data with matrix rank smaller than what it was trained on – hence indicating that the model can find the lowest rank matrix that fits the observed entries. Also please see Fig. A in the attached PDF for results on different (OOD) rank levels, and Fig. 3 in the manuscript for OOD sparsity levels (larger p_mask denotes larger sparsity). We have compared our model in these settings to the nuclear norm minimization solution, to highlight that the BERT model is comparable to, or better than this solution. Please also see the section “OOD performance” in our global response for more information.
>
>
> - *Can the approach be scaled …*
>
> Please see section “Larger matrices” in our global response for addressing this concern.
>
> We hope that our responses have justifiably addressed the reviewer’s concerns and they will consider increasing their score to support the acceptance of our work.

---

> > ### Comment · Reviewer_eRgv · 2024-08-08
> >
> > Thank you for your response. Your replies, as well as your responses to the other reviewers, have resolved my concerns. While I believe that a thorough understanding of the model's grokking mechanism, such as explaining the phenomenon from a dynamic perspective and understanding why the model tends to learn the identity mapping first, and the implications of this mechanism on real-world tasks like eliminating grokking, still has a long way to go, I also think this is a promising area. For example, from a dynamics perspective, you might be able to identify the gradient direction of the model before grokking (the direction of escaping the fixed point) in a suitably simplified model.
> >
> > Based on your response, I raise my score.
> >
> > Additionally, I recommend that you discuss two contemporaneous related works on understanding the grokking phenomenon and analyzing the transformer's mechanism.
> >
> > [1] Grokked Transformers are Implicit Reasoners: A Mechanistic Journey to the Edge of Generalization.
> >
> > [2] Initialization is Critical to Whether Transformers Fit Composite Functions by Inference or Memorizing.

---

### Official Review · Reviewer_1DQa · 2024-07-07

**Soundness:** 3
**Presentation:** 4
**Contribution:** 3
**Rating:** 6
**Confidence:** 3

**Summary:**

The authors study how encoders perform at the task of matrix completion.

They generate synthetic data and train encoders of different sizes to predict masked out tokens.

They observe an interesting behaviour where initially the loss appears to be at a plateau but then drops.

Their hypothesis -which they investigate- is that in the first stage of training the model learns to copy, while in the second stage it learns to predict the masked out tokens.

**Strengths:**

I find the problem formulation interesting, as well as how the observed phenomenon is further investigated by the authors.

**Weaknesses:**

I think the related work could be improved. I think the paper would benefit from a deeper discussion on the experiments and findings of relevant works cited (e.g., 4, 6, 20)

The experimental setting is limited. Synthetic data generation is only carried out using only the uniform distribution. What about other distributions? What about OOD performance?

**Questions:**

Did you consider other distributions?

**Limitations:**

See weaknesses.

---

> ### Author Rebuttal · Authors · 2024-08-07
>
> We thank the reviewer for their feedback, and are glad that they found our problem formulation and analysis interesting!
>
> - *I think the related work could be improved. I think the paper would benefit from a deeper discussion on the experiments and findings of relevant works cited (e.g., 4, 6, 20)*
>
> We are certainly happy to expand on the related work! We will also add the following addendum that discusses references [4, 6, 20] in more detail to the manuscript.
>
> [4] show that in an in–context learning framework, a transformer based model can learn to select the most optimal statistical method for solving the task in the prompt, without explicitly being provided any information about the optimal method (called `in–context algorithm selection’ in their work). We emphasize that our setup is not in–context learning, and is quite distinct from [4] as far as the task being solved is concerned. However, whether the framework of layer-wise in-context gradient descent as in this work can also be shown in our setup is a plausible and open direction for future work (we provide some preliminary investigation in the Probing section in Global response).
>
> In [6], the author shows empirically that a encoder-decoder transformer can be trained to solve various linear algebraic tasks, such as eigendecomposition, SVD, matrix inverse etc. They support their findings by showing that the model generalizes to matrix distributions outside the training distribution to some extent, and that OOD performance can be improved by training on non-Wigner matrices. While many experiments in [6] also show a sudden jump in accuracy (Fig 1,2), they do not analyze why such a sudden jump occurs during optimization. In our work, we analyze the sudden drop and the model before and after it to derive insights into the sudden drop in loss in our training setup.
>
> [20] show that even a small transformer model can be trained to perform arithmetic operations like add, subtract, multiply accurately through appropriate data selection and formatting, and using Chain-of-Thought prompting. They further show that learning addition is connected to rank-2 matrix completion, and that the sudden jump in accuracy with increasing number of observed entries of the matrix is recovered when their model is trained on datasets of different sizes. This is because the size of the dataset for addition can be seen as the number of observed entries of the rank-2 matrix representing the addition table. We point out that while the task in this case is related to matrix completion, ours is a completely different setup, where the sudden drop happens with the number of training steps with each step consisting of 256 low-rank matrices, each with a fixed fraction (p_mask) of observed entries.
>
>
> - *The experimental setting is limited. Synthetic data generation is only carried out using only the uniform distribution. What about other distributions? What about OOD performance?*
>
> Thanks for this suggestion! We have now conducted new experiments to address your comment.
>
> **Changed training distribution.** We now use a standard normal distribution to sample inputs X i.e. X_ij ~ N(0, 1). We find that a sudden drop in the loss occurs in this setting as well, and the MSE before and after the drop matches the values obtained for U[-1, 1] (Fig. D in attached PDF).
>
> **Changed test distribution.** Replacing the distribution of input matrix entries from U[-1, 1] (on which the model is trained) to OOD entries from a Gaussian (mean=0, stdev=0.5) and Laplace distribution (location=0, scale=0.5) does not lead to considerable changes in mean-squared-error (MSE). Specifically, the Gaussian entries yield an MSE of 4e-3 and the Laplace entries yield an MSE of 2e-3 averaged over 1024 samples of 7x7 rank-2 matrix inputs. These results indicate the model has learned a general algorithm to solve the task that is robust to changes in the inputs’ distribution at test time.
>
> **Other interventions.** While we already analyzed OOD generalization performance when varying p_mask (Fig 3 in paper), we have now also conducted experiments by changing the number of rows / columns and rank of the input matrix at test time – see Fig. (A–C) in attached PDF. We find performance is either comparable or better than nuclear norm minimization on the same input, except for the case when we modify the number of columns. This is likely because positional embeddings in the model depend on the column index of the element, and hence (expectedly) changing the number of columns adversely affects model performance.
>
>
>
>
> - *Did you consider other distributions?*
>
> Please see our response to the previous question.
>
> We hope that our responses have justifiably addressed the reviewer’s concerns and they will consider increasing their score to support the acceptance of our work.

---

> > ### Comment · Reviewer_1DQa · 2024-08-13
> >
> > Thanks for your response.

---

### Official Review · Reviewer_h7Ji · 2024-07-13

**Soundness:** 4
**Presentation:** 4
**Contribution:** 3
**Rating:** 7
**Confidence:** 4

**Summary:**

The paper explores the behavior of BERT on matrix completion tasks. The authors show that the model's loss shows a phase transition, where the model switches from copying tokens for filling masked tokens to predicting the masked entries accurately. The authors also conduct probing studies to understand the structure of attention layers, and the hidden representations during the two phases. Overall, the paper takes an important step towards a mechanistic understanding of transformer models, and will be an interesting read for the wider community.

**Strengths:**

The strength of the paper lies in its simplistic exposition of motivation, experimental setup to showcase changing behavior of BERT during training, and the probing studies to explain internal behavior during phase transition. The authors build their motivation from Chen et al. [2024]'s observations on BERT's phase transitions during pretraining on language data, and show the characteristic of such transitions on synthetic matrix completion data. The authors compare the solution to nuclear norm optimization and show that the model outperforms  the candidate algorithm. Finally, with careful probing studies, the authors report the emergence of structure in model's internal representations, which help the model in the mask-token predictions. Overall, the paper takes an important step towards a mechanistic understanding of transformer models, and will be an interesting read for the wider community.

**Weaknesses:**

I have a few questions about the experimental setup and possible interesting followups that the authors may pursue.

- How does convergence of BERT models change with increasing rank of the underlying model? Furthermore, how do results change if the authors mix in matrices of different sizes (say 3x3, 5x5, 7x7, 9x9) but fix rank of the underlying solution? Will the model perform equally well for each of them?

- An interesting ablation would be to check the effect of the model's size on the convergence speed of the model. What happens if the number of attention heads is fixed to $1$ but the width and the depth are changed? More such ablations will help strengthen the understanding of the model.

- In the experiments section "Attention Heads with Structured Mask", how does the removal of each group affect the behavior of the trained model? Are some groups more important than the rest?

- In the "Probing" experiments, "the model tracks information in its intermediate layers and uses it for computation". Is this check only for the row or column number? If so, how does it "only" help the model computation? Is a more fine-grained probing possible, e.g. does the model compute the elements of the low rank decomposition in its embeddings?

- Any discussion on how the results might change for auto-regressive training will be interesting.

**Questions:**

Please check my questions above.

[Post rebuttal]: I have increased my score. The authors' responses have resolved my primary concerns on architecture and matrix ranks. I think this is a timely analysis on training time phase transitions in transformer models in a synthetic setting. The preliminary observations on GPT-2 are very interesting. I hope the authors can include them in the next version.

**Limitations:**

The authors discuss limitations of their work in section 8, and also mention few interesting directions that can be explored by the community as future work.

---

> ### Author Rebuttal · Authors · 2024-08-07
>
> We thank the reviewer for their extensive suggestions, and are happy to know that they found our work important and interesting!
>
> - *How does convergence of BERT …*
>
> Thanks for this suggestion! We have now performed experiments with a 4-layer, 8-head model on rank–1 matrices of size 5x5, 7x7, 9x9 (at each training step, the size is sampled randomly, and then the training data consists of 256 matrices of that size). Interestingly, we find that it matches the intuitive learning order i.e. the sudden drop for 5x5 matrices occurs slightly before that for 7x7 matrices, which in turn is before that for 9x9 matrices (Fig. L in attached PDF). Further, the model performs similarly for 5x5 and 7x7, but is slightly worse for 9x9.
>
> For the question about the effect of rank on convergence, we train our 4 layer 8 head model on 7x7 rank-1 inputs (lower than rank-2 in the manuscript), and find that the sudden drop occurs earlier in this case (Fig. M in attachment). Hence this is an indication that the problem structure affects the rate of convergence, however we leave verifying this for larger matrices and rank to future work.
>
> Finally, in our experiments in the submission, the 12-layer, 12-head model was able to converge on matrices of size upto 15x15, rank–4 (please see Fig. 14 in manuscript for attention maps, and Fig. E in PDF attachment for convergence plot).  We find that going larger than a certain rank for a specific matrix size does not lead to convergence in the usual training setup we report in the paper. This further reinforces the indication above that problem structure affects convergence. Due to limited time for rebuttal and limited computational resources, we are unable to extensively check the largest rank for larger matrix sizes for which our method converges.
>
> (Note that we have modified the input distribution to U[-1.2, 1.2] in rank-1 to match the loss magnitude rank-2 case before sudden drop (~0.06-0.07) for meaningful comparison.)
>
> - *An interesting ablation would be to check …*
>
> Thanks for this suggestion! We have now carried out ablations on model size / architecture to empirically test how it affects convergence. We change model width and depth while keeping other hyperparams the same as the model analyzed in the submitted draft (4–layer, 8 heads, width=768). We also analyze a 1–head, 12–layer model to check for convergence rate when attention layers are altered (please note that 1 head models generally do not converge, likely due to insufficient capacity, but do so for the 12-layer case only). Broadly, our observations are as follows.
>
> **Width (Embedding size)**: we changed the hidden state (embedding) size in the model to 64, 128, 256, 1024, keeping number of layers fixed at 4, and number of heads fixed at 8. For each value of width w, the MLP hidden layer width is 4.w as used in the original w=768 case. We observe (Fig. F in attached PDF) that for embedding size less than 256, the model could not converge to the optimal MSE attained by the larger models.
>
> **Depth (Number of layers)**: we also train a model with depth 2, 6, 8, 12, keeping the number of heads fixed at 8, and width fixed at 768 (Fig. G in attachment; L = number of layers). For depth 2, the model did not converge to optimal MSE and is omitted. We note that for larger models (6, 8, 12), the sudden drop happens earlier than that for depth 4, likely due to larger model capacity in those cases.
>
> **1–head model**: Using a 12–layer, width=768, 1–head model, we find the training converges to MSE ~ 4e-3, with a sudden drop in loss at around step 8000 (Fig J). Interestingly, we find that the attention heads obtained after training (Fig. N; layer 1-12 from left to right) are quite similar to those in 4-layer 8-head models. Possibly, the model exploits residual connections to simulate parallel attention heads within a single layer!
>
>
> - *In the experiments section ...*
>
> Please see section “Structure mask” section in our global response for a response to this concern.
>
> - *In the "Probing" experiments ...*
> This check is for the full masked row corresponding to the element being probed, and not only the row or column number. Specifically, for 768-dimensional hidden state at element (i,j), we map it through a linear model to the 7-dimensional vector X_mask[i, :] where X_mask = X at observed entries, and 0 at masked entries. We hypothesize that this strong correlation indicates that the model stores the masked input in some form in the hidden states, since this information can be probed through a simple linear map. However, we have not claimed that this is the “only” component for the model’s computation.
>
> For probing results on singular vectors (i.e. low rank decomposition), please see section “Probing” in global response.
>
> - *Any discussion on how …*
>
> Great question! We have now carried out preliminary experiments training a GPT model on the matrix completion task. Specifically, the input is now a concatenation of (X_mask, [SEP], X), and the objective is next–token prediction using cross–entropy loss. Due to the changed task structure, we now measure accuracy (instead of MSE) of output tokens at masked, observed, and all entries. Similar to the MLM setting of our main paper, we find that there is an initial plateau in loss (and accuracy), followed by a point of sudden drop in loss! However, the model merely learns to copy the observed entries (Fig. Q in attachment) at this point, and is still struggling to fill in the missing entries. We observe that even this sudden drop corresponds to the model learning specific structure in attention heads for copying (Fig. R in attachment, left: step 200, right: step 600). We are confident that with some hyperparameter and model size tuning, the model will be able to completely solve the task. Due to the limited timeframe for rebuttal, we were unable to finish these experiments, but we promise to include them in the final version of the paper.

---

> > ### Comment · Reviewer_h7Ji · 2024-08-12
> >
> > I thank the authors for their response. I have increased my score. The authors' responses have resolved my primary concerns on architecture and matrix ranks. I think this is a timely analysis on training time phase transitions in transformer models in a synthetic setting. The preliminary observations on GPT-2 are very interesting. I hope the authors can include them in the next version.

---

### Official Review · Reviewer_Kxvs · 2024-07-15

**Soundness:** 3
**Presentation:** 4
**Contribution:** 2
**Rating:** 4
**Confidence:** 4

**Summary:**

The paper applies a BERT-style transformer encoder to do low-rank matrix completion. To frame it as a masked language modeling problem, they restrict the domain of the problem to smaller matrices and discretize the domain.

They find the model can solve the problem. The training dynamics are interesting. Initially the model just copies the input, but after a certain point, it solves the problem.

They analyze many components of before and after this shift: attention heads, activations, and embeddings. They find the behavior of the attention heads changes via visualization and activation patching.

**Strengths:**

It's an easy to read paper with good figures and sound methodology.

The authors anticipate many of the questions and acknowledge limitations.

**Weaknesses:**

The main weakness is how significant the contribution is. They have some interesting results which lead to some hypotheses. In the related work section, they frame the paper as mathematical capabilities of transformers, which is indeed an interesting area. But I don't feel like we've gain much insight into how the transformer does math. The paper feels like preliminary results that could lead to a more interesting paper down the road.

**Questions:**

The authors do a good job of anticipating many questions.

Does this behavior generally to other mathematical tasks?

Can we get any insight about the actually algorithm or implicit optimization procedure?

What is the generalization to larger matrices?

Is there a way to more rigorously verify the hypotheses from the **Attention Heads with Structured Mask** section?

**Limitations:**

The authors acknowledge the limitations in the discussion. This only works on small matrices and is not a generally useful matrix solver.

---

> ### Author Rebuttal · Authors · 2024-08-07
>
> We thank the reviewer for their comments, and are glad to know they found our results interesting!
>
> - *The main weakness is how significant the contribution is. They have some interesting results … how the transformer does math.*
>
> We re–emphasize that our focus is studying the sudden drop in loss from interpretability perspective, and not solving matrix completion itself. We did cite papers on using transformers for mathematical tasks, since our task is also mathematical in nature and wanted to give a review of existing work in this area to the reader. We will definitely consider improving this section to better match the focus of our paper.
>
> We would like to humbly disagree about the significance of our contribution – we emphasize that there are currently no explanations for the phenomenon of sudden loss drops in language modeling. To address this problem, our simplified but rich abstraction offers two concrete observations: (1) the optimization of the model on this task shows an abrupt jump in the model performance and (2) the sudden jump can be analyzed via interpretable evidence that it has learnt the problem structure. These observations allow us to implicate sudden acquisition of task structure by attention layers as the cause behind sudden loss drops. To the best of our knowledge, both our observations and the proposed hypothesis are novel results.
>
> Furthermore, we note that the sudden drop observed in our setting occurs without any changes to the optimization procedure (e.g. step-size, warmup etc) during training – prior work studying such loss drops [1] uses learning rate warmup for 10K steps, while we are able to obtain sudden drop in loss without warmup, and hence have arguably demonstrated a stronger empirical result.
>
>
> [1] https://arxiv.org/abs/2309.07311
>
> - *Does this behavior generally to other mathematical tasks?*
>
> It does! To address your question, we have now extended our experiments to another mathematical task: histogram computation. Specifically, we follow to setup proposed by [2] and train a 2-layer, 2-head GPT model on sequences of the form (x_1, x_2, …, x_32, [SEP] c_1, c_2, …, c_32), where (x_i) are integers in range [1, 16] and (c_i) is the total number of occurrences of x_i in the full sequence. We use such sequences in a next-token prediction (online) training setup for our GPT model, using cross–entropy loss. At test time we prompt the model with incomplete sequences of the form (x_1, x_2, …, x_32, [SEP]) and expect it to complete the sequence with the correct sequence (c_1, c_2, …, c_32).
>
> We find that our observations in matrix completion extend to this setup (Fig. O, P in attached PDF). Specifically,
> there is a sudden drop in training loss, and corresponding increase in accuracy of the model in predicting c_i (Fig. O), and
> attention heads transition from a nearly uniform pattern + attending to [SEP] (step 7000, Fig P - left) to a pattern that attends to various x_i for predicting counts (step 30000, Fig P - right) – as expected for the task.
>
> This shows that such sudden drop in loss, and corresponding emergence of structure in attention heads is not limited to matrix completion and BERT architecture, but generalizes to a different mathematical task of histogram and on a different model (GPT) with autoregressive training, rather than masked language modeling. We are optimistic that such behavior would also extend to other mathematical tasks. We will add these results to the final version of the paper.
>
> We also remark that some experiments in (Fig 1,2) [3] show a sudden increase in accuracy during training, however they do not analyze why that sudden increase occurs, and that is the main focus of our work.
>
>
> [2] https://arxiv.org/abs/2402.03902
> [3] https://openreview.net/forum?id=L2a_bcarHcF
>
> - *Can we get any insight about the actually algorithm or implicit optimization procedure?*
>
> As we show in Section 5, the algorithm learnt by the model shows many interpretable characteristics, indicating that the model learns (1) the problem structure through token and positional embeddings (Fig 7), and (2) how to combine elements at various positions through attention heads (Fig 4,5). Moreover, in our probing experiments (Fig. 6), we find that the model 'stores’ the masked rows of each element in the 3rd and 4th layer, giving insight into the algorithm used by the model. Please also see the section “Probing” in our global response for additional experiments on probing, including a possible hypothesis for implicit optimization in our setup.
>
> Overall, we note that despite our best efforts, we could not precisely nail down the algorithm learned by the model. However, we emphasize this characterization was not the main objective of our paper; instead, we aimed to capture and understand sudden loss drops in the model’s learning dynamics. We did investigate if naively expected hypotheses for how the model might be solving the task check out, e.g., whether the model implicitly learns to perform nuclear norm minimization. Our results were however in the negative, as reported in Fig 3 of the main paper: we found that the model in fact outperforms nuclear norm minimization in terms of MSE.
>
>
> - *What is the generalization to larger matrices?*
>
> Please see section “Larger matrices” in global response.
>
> - *Is there a way to more rigorously verify the hypotheses from the Attention Heads with Structured Mask section?*
>
> Please see section “Structured masks” in our global response – we find that uniform ablation on each category of attention heads leads to increase in L_mask to different extents, largest being for the observed–only heads i.e. (2,2–4), (2,6), (3,2), (3,3), (3,5), (3,1), (3,6).
>
>
> We hope that our responses have justifiably addressed the reviewer’s concerns and they will consider increasing their score to support the acceptance of our work.

---

### Official Review · Reviewer_LwY5 · 2024-07-28

**Soundness:** 3
**Presentation:** 2
**Contribution:** 2
**Rating:** 6
**Confidence:** 3

**Summary:**

The paper examine the ability of the transformer model to solve a low-rank matrix optimization problem. The numerical matrices with missing values are tokenized and flattened as sequences with masked values, and a BERT transformer is trained to predict the original sequence with both masked and unmasked values.  The paper make the following observations through extensive tests:
+ The learning of transformer in matrix completion task exhibits two stages: copying masked values, and sudden transition to learning the underlaying patterns and drastically decreasing loss.
+ During the second, different attention heads exhibits different behaviors correspond to different parts of the input: masks, observed values, position embeddings.
+ The learned token embeddings exhibits clusters between elements from the same row in the original matrix, even when not row/column marker explicitly providing.
These findings aligns with prior studies focusing on masked-language-modeling(MLM) tasks and provides valuable insights into the learning mechanism of transformers.

**Strengths:**

+ The paper proposed on interesting combination by linking low rank matrix completion problem with MLM, potentially enabling the used of many NLP technique to solving the application involving the first problem.
+ The paper performed extensive experiments and examine the learning behaviors of transformers thoroughly across learning stages and important components of transformer.
+ The paper presented sufficient technical details for the experiments implemented such as the loss function and training parameters.

**Weaknesses:**

+ Though using matrix completion as an "abstraction" of MLM is interesting and novel, the author gave little concrete explanation or reference on why insight learned from such setting is transferable to language domain, and what is the benefit over directly analyzing language sequences in terms of gaining interpretation and insights.
+ Despite adapting a new task, the main patterns the paper found in through experiments are similar to prior works.

**Questions:**

+ What is the purpose of training the model to predict all values instead of just the masked ones?
+ What do the two panels in figure represent? The right panel caption appears incorrect and the loss surprisingly goes down as p_mask increases.
+ Is there any consideration for tokenizing the matrix entries, besides mimicking language sequences? Since transformers are increasingly used to processed multimodal data whose input are already embeddings, it would be interesting to see if the same patterns hold for continuous inputs.

**Limitations:**

The authors have addressed societal impact and limitations in the conclusion section.

---

> ### Author Rebuttal · Authors · 2024-08-07
>
> We thank the reviewer for their insightful comments, and are happy to know they found our work interesting!
>
> - *Though using matrix completion as an "abstraction" of MLM is interesting and novel … interpretation and insights.*
>
> We emphasize that our goal while casting matrix completion as an abstraction of MLM was to define a simple, controllable system that captures the phenomenon that we aim to investigate: sudden drops in the loss. First reported by Chen et al. [1] in BERT training, the complexity of natural language disallowed authors to develop precise mechanistic hypotheses for what factors lead to these sudden drops. However, the simplicity of our setting affords us the ability to investigate model internals, showing that the model not only undergoes a change in the algorithm used to solve the given task, but also completely morphs at a mechanistic level (e.g., attention heads specialize to inferring structures relevant to the task). We hypothesize such rapid structure acquisition, specifically driven by attention, drives sudden loss drops. While defining this hypothesis was the goal of this work, given natural language and related practical domains such as code and mathematical problems are filled with rich, structured patterns (e.g., syntactical ones), we expect our hypothesis to hold true therein too. We plan to investigate this more concretely in future work.
>
> We will make sure to add the discussion above to the paper and better clarify why our setup is a faithful abstraction and why we expect our results to transfer across domains.
>
> [1] https://arxiv.org/abs/2309.07311
>
>
> - *Despite adapting a new task, the main patterns the paper found in through experiments are similar to prior works.*
>
> We emphasize that there are currently no explanations for the phenomenon of sudden loss drops in language modeling. To address this problem, our simplified but rich abstraction offers two concrete observations: (1) the optimization of the model on this task shows an abrupt jump in the model performance and (2) the sudden jump can be analyzed via interpretable evidence that it has learnt the problem structure. These observations allow us to implicate sudden acquisition of task structure by attention layers as the cause behind sudden loss drops. To the best of our knowledge, both our observations and the proposed hypothesis are novel results.
>
> Furthermore, we note that the sudden drop observed in our setting occurs without any changes to the optimization procedure (e.g. step-size, warmup etc) during training – prior work studying such loss drops [1] uses learning rate warmup for 10K steps, while we are able to obtain sudden drop in loss without warmup, and hence have arguably demonstrated a stronger empirical result.
>
> [1] https://arxiv.org/abs/2309.07311
>
>
> - *What is the purpose of training the model to predict all values instead of just the masked ones?*
>
> We find that it is difficult to train the model only to predict the masked entries, and predicting all entries substantially aids the training process. We hypothesize that learning to copy the observed entries is helpful for learning to complete the masked entries. We will add this observation to the paper.
>
> - *What do the two panels in figure represent? The right panel caption appears incorrect and the loss surprisingly goes down as p_mask increases.*
>
> We note that the caption is indeed correct. Specifically, the right panel in Fig. 3 compares the nuclear norm of the solutions obtained by BERT and nuclear-norm minimization – in this case, as p_mask increases, MSE displayed in the left panel goes up, as expected. We apologize for the lack of clarity though, and promise to make the caption clearer in the final version of the paper.
>
>
>
> - *Is there any consideration for tokenizing the matrix entries, besides mimicking language sequences? Since transformers are … if the same patterns hold for continuous inputs.*
>
>
> Indeed, the motivation behind tokenization is to keep the setup similar to language sequences, where BERT has been shown to perform well empirically. Importantly, as we show in Fig. 7, the model can learn the token embeddings representing the real values with expected structure (principal components separated by sign of the input, and continuously varying with magnitude of input). Hence it should not significantly affect the results if we use specialized embeddings instead of our approach; extending our work to other embedding methods is an interesting avenue for future work.

---

### Author Rebuttal · Authors · 2024-08-07

We thank the reviewers for their detailed feedback and are excited to see that they find our work interesting! To address raised questions, we have performed several new experiments, as described below. We will add these results to the final version of the paper.

**Larger matrices.** We show that our setup extends to matrices of size 15x15, rank-4 with a 12–layer, 12–head model with similar training dynamics (please see Fig. E in PDF attachment) and final attention head maps (please see Fig. 14 in manuscript). Extending to very large matrices is limited by the context length of transformers and compute resources; though we are optimistic that such results should hold in general for larger matrices. We would also like to re–emphasize that our main focus is not solving matrix completion through transformers, but study transformers in a mathematical setup to (i) model the sudden drop in loss and (ii) interpret the model behavior to understand why this drop occurs.


**Probing.** We probe for the true matrix element at each position in each layer; i.e., map the 768-dimensional hidden states at position (i,j) for input X_mask, to the real value X_{ij} (Fig. H in attached PDF). Interestingly, we find that the MSE decreases at an exponential rate for masked positions. We conjecture that this might be related to the idea that each layer computation of the transformer implicitly corresponds to a step of some optimization algorithm (e.g. gradient descent [1, 2]), however we could not empirically verify that this is the case for our setup. We leave it to future work to confirm if this conjecture holds true in our case.

We also attempted to probe the singular vectors of the ground truth matrix in the hidden states of the model. Concretely, we map the 768-dimensional hidden states at different layers through a linear model for a given input matrix X, to the 7-dimensional first left singular vector u of X (i.e. if X = U.S.V^T is the SVD of X, then u = first column of U). We found that the MSE is too large (~0.14), and the cosine similarity of the estimated vector and the true vector is near random (<0.3 i.e., less than the average cosine similarity of 7-dim vectors sampled from N(0, I)); please see Fig. (I) in the PDF attachment for these results for all layers. Hence, it is not immediately clear that the model has some information about the singular vectors of the input matrix stored in its hidden states, that is recoverable through linear probes. Similar finding holds for other singular vectors i.e second column of U, and first 2 columns of V.

[1] https://arxiv.org/abs/2306.04637

[2] https://proceedings.mlr.press/v202/von-oswald23a.html

**Structured mask.** We find that uniform ablation on each category of attention heads in Section "Attention Heads with Structured Mask" leads to increase in L_mask to different extents. The maximum effect of ablations is in the observed heads category ((2,2–4), (2,6), (3,2), (3,3), (3,5), (3,1), (3,6)), quantified by measuring the ratio of MSE with ablations to MSE without ablations. The first layer attention heads that are claimed to possibly “process positional and token embeddings” in row 5 in the table, also affect the model output to a larger extent. This rigorously verifies the hypothesis that these attention head groups causally affect the model output.


**Effect of input distribution**

- Training distribution.
For training the model, we replace the distribution U[-1, 1] by a standard normal distribution (i.e. X_ij ~ N(0, 1) for input X). We can indeed recover the sudden drop in loss, where the MSE before and after the drop matches the values obtained for U[-1, 1]. Please see Fig. D in the attachment for reference.


- Test distribution.
For test time OOD performance, we find that replacing the distribution of input matrix entries from U[-1, 1] (on which the model is trained), with Gaussian (mean=0, stdev=0.5) and Laplace (location=0, scale=0.5) does not lead to any considerable change in mean-squared-error (MSE). Specifically, the Gaussian entries yield an MSE of 4e-3 and the Laplace entries yield an MSE of 2e-3 averaged over 1024 samples of 7x7 rank-2 matrix inputs.

Please note that we already check OOD performance when varying p_mask (Fig 3 in manuscript). Further, we also check OOD performance by changing the number of rows / columns and rank of the input matrix – as observed in Fig. (A–C) in attached PDF, BERT performance is either comparable or better than nuclear norm minimization on the same input, except for the case when we modify the number of columns. This is because positional embeddings in our model depend on the column index of the element, and hence (expectedly) changing the number of columns adversely affects model performance. The overall idea is that as long as the input matrix entries do not exceed beyond a certain threshold in magnitude, the model can solve matrix completion. This is because the model has learnt token embedding representations for a fixed range of input values (as defined by U[-1, 1]). It is unexpected to generalize to larger entries that the model does not see during training, since the model has no knowledge of how tokens for larger entries are represented in the embedding space. This also matches the observations in  Sec 5 [3], where the author shows that changing the variance of test distribution significantly affects OOD performance of the model.

[3] https://openreview.net/forum?id=L2a_bcarHcF

---

### Decision · Program_Chairs · 2024-09-25

**Decision:**

Accept (poster)

**Comment:**

This submission has received scores of 6, 4, 7, 6, 5, indicating a weak acceptance of this submission. The reviewers generally find the paper interesting and well-presented, with a novel approach to studying the behavior of transformer models on a simplified matrix completion task. However, some concerns remain about the significance of the contributions and the generalizability of the findings. Still, the reviewers' scores and comments, along with the authors' detailed and responsive rebuttal, suggest that the paper's strengths outweigh its weaknesses. While there is room for further exploration and deeper mechanistic understanding, the paper makes a valuable contribution to the field and warrants acceptance.

The authors are encouraged to address the reviewers' concerns in the final version by expanding the discussion on the significance and generalizability of their findings, exploring the possibility of extending their analysis to larger-scale tasks and different model architectures, and providing a deeper theoretical or mechanistic understanding of the observed phenomena.

Strengths:
- Novelty and clarity: The paper presents a novel and well-executed approach to studying transformer behavior on a simplified task, making it easy to follow and understand.
- Interesting observations: The phase transition phenomenon and the analysis of attention heads and embeddings provide valuable insights into the model's learning dynamics.
- Thorough experimentation: The authors conduct extensive experiments and provide detailed descriptions of their methodology.

Areas to Improve:
- Significance of contributions: Some reviewers question the overall impact and generalizability of the findings, particularly regarding the transferability of insights to the language domain and larger-scale tasks.
- Depth of mechanistic understanding: While the paper offers valuable observations, a deeper theoretical or mechanistic analysis of the observed phenomena is desired.
- Generalizability: The experiments are primarily conducted on small matrices, raising concerns about the scalability and applicability of the findings to more complex scenarios.